# 1-D velocity structure modelling of the Earth's Crust in the NW Dinarides

Gregor Rajh[1], Josip Stipčević[2], Mladen Živčić[3], Marijan Herak[2], Andrej Gosar[1,3], and the AlpArray Working Group[+]

[1]University of Ljubljana, Faculty of Natural Sciences and Engineering, Ljubljana, Slovenia
[2]University of Zagreb, Faculty of Science, Geophysics Department, Zagreb, Croatia
[3]Slovenian Environment Agency, Seismology Office, Ljubljana, Slovenia
[+]The complete member list of the AlpArray Working Group is at the end of the paper.

*Correspondence to*: Gregor Rajh (gregor.rajh@ntf.uni-lj.si)

**Abstract.** The studied area of the NW Dinarides is located in the NE corner of the Adriatic microplate and is bordered by the Adriatic foreland, the Southern Alps, and the Pannonian basin. Its complex crustal structure is the result of interactions among different tectonic units, the most important of which are the Eurasian plate and the Adriatic microplate. Despite numerous seismic studies in this tectonically complex area, there is still a need for a detailed, small-scale study focusing mainly on the upper, brittle part of the crust. In this work, we investigated the velocity structure of the crust with 1-D simultaneous hypocenter-velocity inversion using routinely picked P- and S-wave arrival times. Most of the models computed in the combined P and S inversion converged to a stable solution in the depth range between 0 and 26 km. We further evaluated the inversion results with hypocenter shift tests, high and low velocity tests, and relocations. This helped us to select the best performing velocity model for the entire study area. Based on these results and the seismicity distribution, we divided the study area into three subregions, reselected earthquakes and stations, and performed the combined P and S inversion for each subregion separately to gain better insight into the crustal structure. In the eastern subregion, the P velocities in the upper 8 km of the crust are lower compared to the regional velocities and the velocities of the other two subregions. The P velocities between 8 and 23 km depth are otherwise very similar for all three models. Conversely, the S velocities between 2 and 23 km depth are highest in the eastern subregion. The north-western and south-western subregions are very similar in terms of the crustal structure between 0 and 23 km depth, with slightly higher P velocities and lower S velocities in the SW subregion. High $v_P/v_S$ values were obtained for the layers between 0 and 4 km depth. Below that, no major deviations of $v_P/v_S$ in the regional model from the value of 1.73 are observed, but in each subregion we can clearly distinguish two zones separated by a decrease in $v_P/v_S$ at 16 km depth. Compared to the model currently used by the Slovenian Environment Agency to locate earthquakes, the obtained velocity models show higher velocities and agree very well with some of the previous studies. In addition to the general structural implications and the potential to improve the results of seismic tomography, the new 1-D P and S velocity models can also be used for reliable routine earthquake location and for detecting systematic travel time errors in seismological bulletins.

# 1 Introduction

The study area of the NW Dinarides lies at the north-eastern corner of the Adriatic microplate and is bounded by the Southern Alps to the north, the Pannonian basin to the east, and the Adriatic foreland to the west, thus representing an important junction between these units (Fig. 1). The evolution of the Dinarides is tied to the ongoing collision between the Eurasian plate (Eurasia) and the Adriatic microplate (Adria), which began in the late Cretaceous (Tari, 2002; Handy et al., 2010; Ustaszewski et al., 2010; Handy et al., 2015).

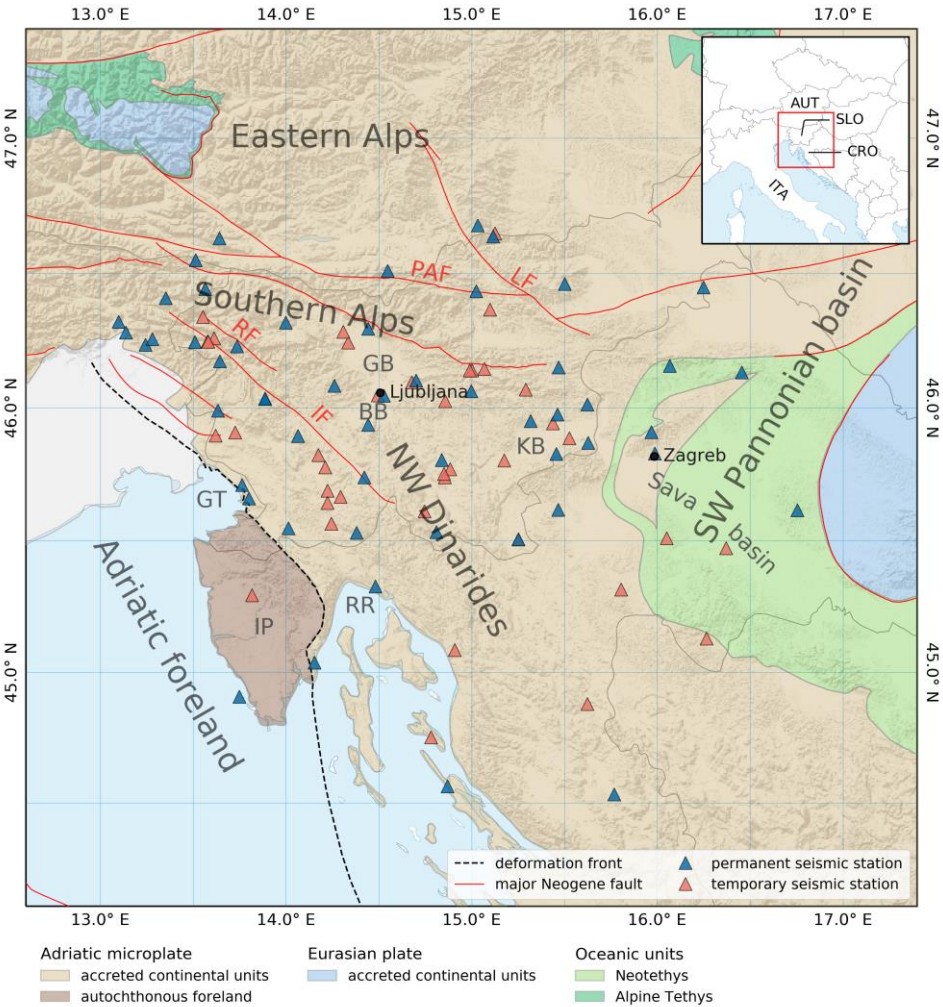

**Figure 1: Map of the study area. Seismic stations used in this study are shown on top of the regional tectonic map and major Neogene faults (adapted from Schmid et al., 2008). Black dashed line represents the current main deformation front of the Dinarides. GT, Gulf of Trieste; IP, Istra peninsula; RR, Rijeka region; GB, Gorenjska basin; BB, Barje basin; KB, Krško basin; IF, Idrija fault; PAF, Periadriatic fault; RF, Ravne fault; LF, Labot fault; AUT, Austria; CRO, Croatia; ITA, Italy; SLO, Slovenia. Shaded relief is shown in the background (Esri, USGS, NOAA).**

First 3-D compressional (P) wave velocity model in this area was obtained with local earthquake tomography (LET) study done by Michelini et al. (1998). It revealed two areas of distinct high and low velocities in western and eastern Slovenia, which were interpreted as the upper crustal expression of the ongoing convergence between the Adria and the Eurasia. The authors also proposed a relocation study using the 3-D velocity model to map active faults and trends in seismicity. This was

partly realised by a study that focused on the Idrija fault system in western Slovenia (Vičič et al., 2019). Its authors were able to constrain the geometry of each fault by relocating seismicity with the regional 3-D shear (S) wave velocity model of Guidarelli el al. (2017) and a constant P-wave to S-wave velocity ratio ($v_P/v_S$). The model of Guidarelli et al. (2017) was obtained with ambient seismic noise tomography and shows distinct lateral change in the crustal structure under western Slovenia. This was interpreted as a transformation from a uniform to a more variable crustal structure across the bounding

strike-slip Idrija fault, indicating the transition between the Dinarides and Pannonian basin units. Recently, Kapuralić et al. (2019) computed a 3-D P-wave velocity model from LET and used these results to constrain the relationship between the crust and uppermost mantle at the junction between the Dinarides and the Pannonian basin. Their findings show significant changes in the crustal structure at the transition zone between the NW Dinarides and the Pannonian basin and map several zones of higher seismic velocity in the NW Dinarides crust. As opposed to the model of Guidarelli et al. (2017), this 3-D

velocity model shows no obvious crustal signature of the dividing Idrija fault. The 3-D velocity models of Bressan et al. (2012) provided insights into the upper crustal structure of NE Italy and cover the NW corner of our study area. The models show strong variations in P-wave velocity and $v_P/v_S$ related to lithological heterogeneities and variable degree of fracturing caused by several different tectonic phases. The surface wave dispersion study in Slovenia by Živčić et al. (2000) showed a 4-6 km thick layer with S-wave velocities between 2.75 and 3.00 km s$^{-1}$ above a 7-9 km thick layer with S-wave velocities

between 3.00 and 3.30 km s$^{-1}$. The velocity of the underlying layer was found to be lower in eastern Slovenia. Their results also suggest comparatively higher velocities in deeper parts of the upper crust in western Slovenia. The most recent 3-D model of the region was constructed by Magrin & Rossi (2020) by critically selecting and integrating all available information on the depth of the primary interfaces and the physical properties of the crust. Using this model, they were able to tie the spatial distribution of the seismicity in the northern part of the Adria to the sharp changes in various physical

parameters in the crust. Active seismic investigations of Brückl et al. (2007) and Šumanovac et al. (2009) provided important insights into the crustal structure along profiles crossing the Alps and Dinarides, constraining P-wave velocities and the depth of the Mohorovičić discontinuity (Moho). Direct comparison between P- and S-wave velocity models should be done with care due to the highly variable average $v_P/v_S$ values in the region (Behm, 2009; Stipčević et al., 2020). The latest receiver functions study applied to the Dinarides and the surrounding area (Stipčević et al., 2020) showed the transition from

the thick Dinaric to the thinner Pannonian crust and indicated that the earthquake depths generally follow the crustal thickness.

Despite the numerous investigations that completely or partially covered the study area, the details of the upper crustal structure remained unresolved. Moreover, the 3-D velocity models covering the study area show markedly different and rapid lateral velocity variations in the upper crust. For these reasons, there is still a need for a detailed, small-scale study focusing mainly on the upper, brittle part of the crust. Therefore, our goal is to investigate the velocity structure of the crust using the concept of a minimum one-dimensional (1-D) velocity model. The minimum 1-D velocity model is computed by simultaneous inversion for hypocenter and velocity parameters (coupled hypocenter-velocity problem) and represents the best fit to the observed travel time data in the least-squares sense. This iterative approach is necessary because of the strong coupling between hypocenter and velocity parameters (Kissling, 1988; Kissling et al., 1994). If obtained properly, the minimum 1-D velocity model can be used to calculate accurate earthquake locations (e.g., Husen et al., 1999) and to detect systematic errors in travel time data (e.g., Maurer et al., 2010), especially when computed at a smaller scale (Husen et al., 2011). Station delays computed as part of the minimum 1-D velocity model allow identification of major geological and tectonic features or trends. The minimum 1-D velocity model is also essential for 3-D velocity modelling in LET, where it is commonly used as an initial model in inversion (e.g., Kissling, 1988; Kissling et al., 1994; Haslinger et al., 1999; Diehl et al., 2009). Using a minimum 1-D velocity model as the initial model in LET can greatly reduce inversion artefacts in a final 3-D velocity model and improve error estimates (Kissling et al., 1994).

Most of the seismicity in the study area has been located with the synthetic 1-D velocity model (routinely used 1-D velocity model, R1D from now on) aggregated mainly from the results of Michelini et al. (1998) and Živčić et al. (2000). Compared to today's situation, the results of these studies were obtained with a relatively small amount of data. Seismic station coverage in the area improved significantly with the gradual modernization of the Seismic network of the Republic of Slovenia (SNRS) between 2001 and 2008 (Vidrih et al., 2006; Jesenko & Živčić, 2018) and the deployment of additional seismic stations in Croatia within the VELEBIT project (2015-2019), and during 2015 and 2016 as part of the AlpArray project (Molinari et al., 2016). With better station coverage and smaller epicentral distances, we are now able to sample the upper crustal structure more densely, and therefore calculate more accurate upper-crustal velocity models. Furthermore, studying spatial distribution of the relocated seismicity allows us to put additional constraints on the crustal structure and the processes driving the seismicity itself.

## 2 Tectonic setting and crustal structure

The tectonic evolution of the study area is closely related to the dynamics of the Adria (e.g., Anderson & Jackson, 1987). The subduction processes associated with the closure of the Neotethys ocean started in the Jurassic (Pamić et al., 1998; Tari, 2002; Schmid et al., 2008) and led to the continental collision between the Adria, which at that time detached from the African plate (Schmid et al., 2008), and the Eurasia in the late Cretaceous (Tari, 2002; Handy et al., 2010; Ustaszewski et al., 2010; Handy et al., 2015). The collisional processes that occurred along the northern (e.g., Kissling et al., 2006) and western

(e.g., Vignaroli et al., 2008) margins of the Adria, gave rise to the Alps and the Apennines, respectively. Along the eastern margin, the collisional process started after the oceanic part of the Adria was consumed in the subduction, leading to the formation of the thrust sheets and the ophiolitic units of the Neotethys (Schmid et al., 2008). The subduction ceased in the early Paleogene (Pamić et al., 1998; Schmid et al., 2008) and the deformation front began to migrate south-westward (Tari, 2002; Korbar, 2009; Ustaszewski et al., 2010; Handy et al., 2015). The peak of still ongoing deformation lasted until the

early Oligocene and was expressed by the foreland directed thrusting (Pamić et al., 1998; Tari, 2002; Schmid et al., 2008; Placer et al., 2010), which strongly deformed the upper parts of the Adria crust (Schmid et al., 2008; Korbar, 2009). At the same time, the continental part of the Adria began to underthrust the Dinarides (Tari, 2002; Placer et al., 2010). In addition, the movement of the Adria was responsible for the late Oligocene-Miocene south verging thrusting in the Southern Alps (Schmid et al., 2004; Handy et al., 2010; Handy et al., 2015) and the lateral extrusion of the Eastern Alps along the ENE-

WSW striking Periadriatic fault, which separates the Southern Alps from the Eastern Alps (Fodor et al., 1998). An important factor in the process of lateral extrusion of the Eastern Alps was the extension of the area behind the retreating subduction zone in the Carpathians (Ratschbacher, 1991a, 1991b; Horváth & Cloetingh, 1996), which led to rifting and subsidence, resulting in the formation of the Pannonian basin (Fodor et al., 1998). In the late Miocene to early Pliocene (Márton et al., 2003; Márton, 2006), a sustained counterclockwise rotation of the Adria began (Anderson & Jackson, 1987; Battaglia et al.,

2004; Grenerczy et al., 2005; Weber et al., 2010) and the end of subduction in the Carpathians (Horváth & Cloetingh, 1996) led to transpressive reactivation of the former extensional structures in the Pannonian basin (Horváth & Cloetingh, 1996; Fodor et al., 1998; Fodor et al., 1999; Tari, 2002, Grenerczy et al., 2005; Ustaszewski et al., 2010) and to transpressive to purely strike-slip deformation along the zone of steep, NW-SE striking faults in the Dinarides and the Southern Alps (Picha, 2002; Placer et al., 2010; Vičič et al., 2019). Currently, shortening in the area is more strongly absorbed on the southern front

of the Eastern Alps in the area of the Sava fault, with movement increasingly turning eastwards towards the Pannonian basin (Métois et al., 2015).

Crustal thickness in the NW Dinarides, has been recently constrained by many different studies (Brückl et al., 2007; Behm et al., 2007; Šumanovac et al., 2009; Stipčević et al., 2011; Guidarelli et al., 2017, Kapuralić et al., 2019; Stipčević et al.,

2020). It varies from about 38 to 45 km under the External Dinarides, slightly thickening towards the Alps and thinning to about 30 km in the Adriatic foreland and 25 km in the Pannonian basin. A similar pattern was observed for the lithosphere thickness in the same area (Belinić et al., 2018). The underthrusting of the Adria resulted in two-layered thickened crust in the External Dinarides. The thinner crust in the Adriatic foreland is associated with the undeformed parts of the Adria. The extension in the late Oligocene and early Miocene, which caused crustal thinning in the Pannonian basin, is most likely

responsible for relatively low P-wave seismic velocities in the upper and middle crust under the transition zone from the Southern Alps and the Dinarides to the Pannonian basin. The thinned crust in contact with the Adria in this transition zone belongs to the Pannonian fragment. The junction between these two units appears as a 10 km jump in Moho depth, probably a result of the crustal thinning (Brückl et al., 2007; Brückl et al., 2010).

Throughout the study area the seismicity is mostly constrained to the upper crust (Herak et al., 1996, Slovenian Environment Agency, 2019). Several strong historical and instrumentally recorded earthquakes occurred in this region. The strongest historical earthquake with estimated magnitude of about $M_W$=6.8 and a maximum estimated intensity of X EMS-98 occurred in 1511 on the Idrija fault in western Slovenia (Vidrih & Ribičič, 2004; Fitzko et al., 2005; Cecić & Jocif, 2011). The Rijeka region was hit by four damaging earthquakes between 1750 and 1904 with maximum intensity estimates from VI to VIII

MSK (Herak et al., 2017; Herak et al., 2018). The strongest historical earthquake near Zagreb occurred in 1880 with a maximum intensity of VIII MCS (Herak et al., 1996). Shortly after, two destructive earthquakes occurred in Slovenia. In 1895, an earthquake near Ljubljana (central Slovenia) occurred with $M_W$ 6.0 (VIII-IX EMS-98) (Lapajne, 1989; Tiberi et al., 2018) and in 1917, an earthquake with $M_W$ 5.6 (VIII EMS-98) struck the Krško basin (Lapajne, 1989; Cecić et al., 2018). Recently, two strong earthquakes occurred on the Ravne fault in north-western Slovenia. The first one in 1998 with $M_W$ 5.6

and a maximum intensity of VII-VIII EMS-98 (Zupančič et al., 2001) was followed by an earthquake with $M_W$ 5.2 (VII EMS-98) on 12 July 2004 (Vidrih & Ribičič, 2004). A review of the seismological, geological, and seismotectonic studies related to both earthquakes was given by Gosar (2019a, 2019b). Detailed LET study with aftershock sequences of these two main events was performed by Bressan et al. (2009), linking the physical properties of the crust along the Ravne fault with the spatial seismicity distribution and different types of focal mechanisms. The most recent damaging events in this area

occurred in Croatia near Zagreb ($M_W$ 5.4; Atalić et al., 2021) and Petrinja ($M_W$ 6.4; Tondi et al., 2021) in 2020.

**3 Data**

The seismological bulletin of the Slovenian Environment Agency (ARSO), consisting of 7,733 local earthquakes with $M_L$ of at least 1.0 that occurred between 2004 and 2018, served as a starting point for this study. The earthquakes are routinely analysed by ARSO and cover the entire territory of Slovenia and its surroundings. Their locations were determined with the

HYPOCENTER program (Lienert & Havskov, 1995) using P and S arrival times and the routine 1-D velocity model. Mining blasts are removed from the main catalogue and are used as an independent dataset for testing. The arrival time picks (arrivals) in the seismological bulletin were grouped into six uncertainty classes based on the uncertainty intervals subjectively determined by the analysts, as shown in Table S1. In our dataset, the best estimated first arrivals (classes 0, 1, 2) dominate and only a small number of arrivals belong to uncertainty classes of 3 and 4. For our study, we considered only the

arrivals that belong to uncertainty classes of 0, 1, and 2.

Most earthquakes in the study area are confined to depths between 1.1 km and 18.3 km (5th and 95th percentiles). The earthquake with $M_L$ 4.9 ($M_W$ 5.2) that occurred on 12 July 2004 (Vidrih & Ribičič, 2004) is the strongest earthquake in our dataset and one of the few that exceeded $M_L$ 4.0. Earthquakes of the lowest magnitude considered ($M_L$ 1.0) had on average 9

P and 8 S arrivals, which is sufficient for a good location estimate. Moreover, the arrival times of these smaller earthquakes were still reliably picked (uncertainty class ≤ 2) at maximum average epicentral distance of about 84 km.

The study area is densely populated with seismic stations (Fig. 1). The arrival times were picked mainly at seismic stations of the Seismic Network of the Republic of Slovenia (Slovenian Environment Agency, 2001) together with seismic stations
belonging to other seismic networks and temporary seismic arrays in the region (Zentralanstalt für Meteorologie und Geodynamik, 1987; MedNet Project Partner Institutions, 1990; University of Zagreb, 2001; OGS, 2002; INGV Seismological Data Centre, 2006; AlpArray Seismic Network, 2015; OGS, 2016). The Seismic Network of the Republic of Slovenia (SNRS) was gradually modernised between 2001 and 2008 and currently consists of 26 permanent stations (Vidrih et al., 2006; Jesenko & Živčić, 2018). During the last 16 years, many temporary stations have also been in operation in
Slovenia. In recent years, some additional seismic stations have been installed as part of the VELEBIT and AlpArray projects (Molinari et al., 2016), filling the gaps between the permanent stations of the Croatian Seismic Network (CR). Some seismic stations located in Austria and Italy were also used to cover the periphery of our study area. The seismic network operating in north-eastern Italy is managed by the Italian National Institute of Oceanography and Applied Geophysics – OGS and is described in detail in Bragato et al. (2021).

**4 Method**

Observations of seismic phase arrival times can be used to investigate seismic velocity structure of Earth's interior. Arrival time of a wave ($T_{ij}$) generated by an earthquake ($i$) and observed at a station ($j$) is a nonlinear function of station coordinates ($s_j$), hypocenter parameters ($h_i$), and velocity model parameters ($m$). This function can be approximated with a Taylor series expansion about the points in a hypocenter and a velocity model solution space ($h_i^0$, $m^0$). By only keeping linear terms we
obtain its linearised form

$$T_{ij} = T_{ij}^0 + \sum_{k=1}^4 \left[\frac{\partial T_{ij}}{\partial h_{ki}}\right]_{h_i^0, m^0} \Delta h_{ki} + \sum_{k=1}^p \left[\frac{\partial T_{ij}}{\partial m_k}\right]_{h_i^0, m^0} \Delta m_k + e \,, \tag{1}$$

which relates small changes in arrival time to small changes in the hypocenter and the velocity model parameters. The third term is summed over the total number of velocity model parameters ($p$). The error term ($e$) contains arrival time errors caused by the approximation and errors in calculated and observed arrival times.


By estimating (predicting) hypocenter and velocity model parameters, we can calculate arrival time ($T_{ij}^0$) of an earthquake phase, and all partial derivatives in Eq. 1. We do this numerically by tracing rays for predicted hypocenter parameters through predicted velocity structure (e.g., Crosson, 1976; Kissling, 1988). The difference between the calculated and observed arrival time can be expressed as an arrival or travel time residual, which is related to the perturbations (corrections) in the hypocenter and velocity model parameters, $\Delta h_{ki}$ and $\Delta m_k$, respectively. For $I$ earthquakes, each observed at $J$ stations,

we obtain a system of $N = I \times J$ linear equations, which we solve by minimizing the misfit (residual) to the data with the damped least squares approach (e.g., Crosson, 1976; Aki et al., 1977; Kissling, 1988). Because we are solving simultaneously for hypocenter and velocity parameters, this inverse problem is known as the coupled hypocenter-velocity problem. Since the system of equations which we are solving is not a true linear system, the hypocenter and velocity model perturbations must be small. Therefore, an initial estimate of the unknown parameters must be sufficiently close to the correct solution and the inversion performed iteratively by adjusting hypocenter and velocity model parameters in each step (Crosson, 1976).

The result of the coupled hypocenter-velocity problem described above is the velocity model (velocities and possibly station delays) and the revised hypocenter parameters. The resulting model minimizes the travel time residuals and is referred to as the minimum 1-D velocity model in the case of the 1-D parameterization. The layer velocities of a 1-D velocity model approximate the average velocity of a 3-D velocity model in the same depth interval. The construction of a minimum 1-D model is a trial-and-error process that requires careful selection of only high-quality data and rigorous evaluation of the results (Kissling et al., 1994).

## 5 1-D velocity modelling

### 5.1 Initial dataset

To sufficiently sample the solution space, five different initial 1-D P-wave velocity models (Fig. 2) were used as input to the inversion. Three of these were derived from the independent studies (Brückl et al., 2007; Šumanovac et al., 2009) and from the synthetic 1-D velocity model routinely used by ARSO to locate earthquakes (R1D). Two initial models with low and high velocity values were also included in the inversion procedure. They were subjectively defined to roughly envelop the lowest and highest velocity values of the three initial models derived from independent studies with an average buffer of about 0.15 km s$^{-1}$, while keeping the inversion stable. By using several different models, we were able to better sample the solution space and test the dependence of our solution on an initial model. To define the layered structure, we started with thicker layers and thinned them at more densely sampled depth intervals, paying close attention to the change in the RMS residual and the convergence of the models. The difference in thickness between adjacent layers was kept as small as possible to ensure stability during the inversion. The surface layer (above 0 km) is used to account for station elevations, and its velocity generally shows stronger coupling with station delays.

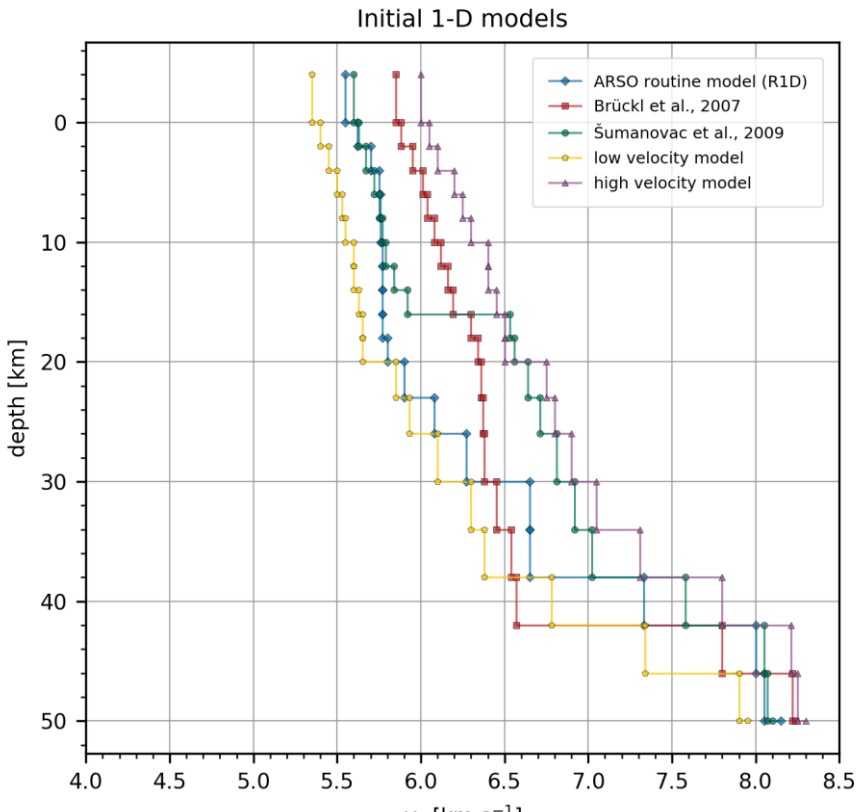

**Figure 2: Initial 1-D models derived from the independent studies (blue, red, green). The low (yellow) and high (purple) velocity models roughly envelop the three initial models derived from the independent studies and are used to further sample the solution space.**

The goal of the earthquake selection procedure is to select a high-quality earthquake dataset that is uniformly distributed over the volume under study and has the highest number of good first arrivals. Routinely determined hypocenter parameters were used and we kept only the first arrivals with an uncertainty class of 2 or better picked at the selected seismic stations (Fig. 1). Earthquakes with a depth of 0 km, a maximum azimuthal gap greater than 160°, and RMS residual of more than 0.5 s were removed. After several tests, the studied area was tessellated into square cells of 10 km (Fig. 3). For each cell, events were sorted by their parameters and iteratively selected to obtain the most diverse depth distribution possible and avoid clustering. This was achieved by setting the minimum vertical distance between earthquakes within a single cell to 2 km, a value determined by a trial-and-error approach based on the final number of earthquakes selected. Earthquakes in each cell were hierarchically sorted by (in descending order of importance) a total number of travel times with an uncertainty class of 0, a total number of travel times with an uncertainty class of 1, a total number of all travel times, an azimuthal gap, a

magnitude, and a total number of stations with readings. The first earthquake from the sorted list was selected and then the others followed iteratively according to the minimum depth distance.


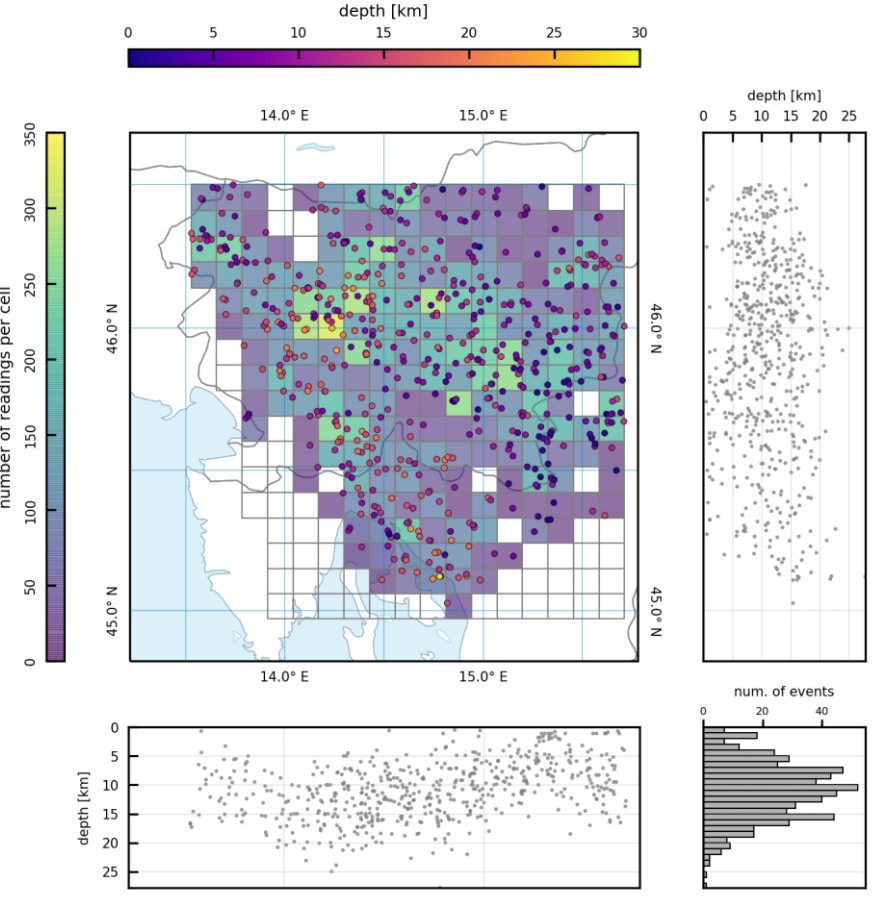

**Figure 3: Earthquake dataset selected for the combined P and S inversion. Square 10 km cells shown on the main map were used to select earthquakes. Colour of each cell represents the total number of P and S readings per cell. The right and bottom panels show the hypocenters of earthquakes projected on N-S and W-E oriented profiles, respectively. The histogram in the lower right**
**corner shows the number of earthquakes in 1 km depth bins for the whole study area.**

Using the earthquake selection procedure described above, we obtained a high-quality dataset needed for each type of inversion (Table 1). For the independent inversion of P velocity parameters (P-only inversion), 634 earthquakes with at least 10 remaining first P arrivals were selected, a total of 15,742 readings, and a maximum epicentral distance of 266 km. Of
these, 14,848 were manually picked as Pg phases, while 423 were picked as Pn phases. The second dataset used for the independent inversion of S velocity parameters (S-only inversion) contains 521 earthquakes with at least 8 remaining first S arrivals and a total of 7,914 readings with a maximum epicentral distance of 260 km, 7562 Sg phases, and 126 Sn phases.

The final dataset for the inversion of both P and S velocity parameters (combined P and S inversion) consists of 582 earthquakes (Fig. 3) with at least 10 remaining first P and 5 first S arrivals. This dataset contains 13,034 readings of first P

and 10,134 readings of first S arrivals with a maximum epicentral distance of 260 km. Of these, 12,346 were manually picked as Pg phases, 325 as Pn phases, while 9,600 were picked as Sg phases and 242 as Sn phases. Epicentral ray coverage was determined for each dataset by connecting earthquake station pairs with great circles and counting rays intersecting 10 km grid cells. An example for P and S dataset is shown in Fig. 4. The earthquake selection grid was truncated in places where seismicity is sparse (e.g., the Istra peninsula). Its extent was also limited by the spatial extent of the earthquakes in the

seismological bulletin. We also made sure that the selected earthquakes were entirely within the area defined by the seismic stations.

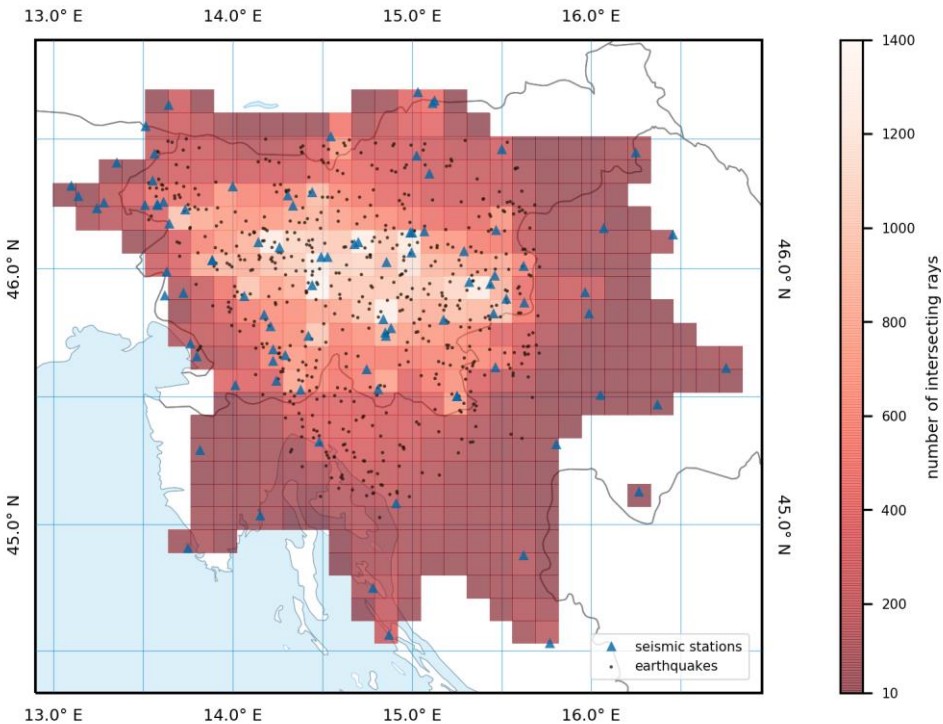

**Figure 4: Number of great-circle rays intersecting square 10 km grid elements and connecting earthquake-station pairs for the**
**earthquakes shown in Fig. 3. Grid elements with less than 10 intersecting rays are not shown.**

### 5.2 Modelling process

To compute 1-D velocity models, we used the VELEST code (Kissling et al., 1994; version 4.5) and followed the guidelines for computing a minimum 1-D velocity model from Kissling et al. (1994), Husen et al. (2011), and the VELEST user manual (Kissling, 1995). The VELEST code has been improved through the efforts of many authors and has become very versatile

and robust. It also allows the calculation of station delays, which enter the inversion as unknown velocity model parameters

and are thus part of the 1-D velocity model (Kissling, 1988). In general, the computation of a 1-D velocity model was performed in two runs. In the first run, the hypocenter parameters were computed at each iteration, while the velocity model parameters were adjusted along with the hypocenter parameters at every other iteration. This approach was necessary because we performed a separate inversion for each initial 1-D velocity model with a set of routinely determined initial

hypocenter parameters. In addition, this run allows for large perturbations in velocities. We set the damping to 0.01 for the hypocenter parameters and station delays, and to 0.10 for the velocity parameters, as suggested by Kissling et al. (1994). By increasing the damping of the velocity parameters to 0.20 in the S-only inversion, we were able to avoid the instabilities that terminated the iterative process in the first simultaneous hypocenter-velocity iteration step. In the second run, the hypocenter parameters and velocity model obtained in the first run were used as input, the station delays were set to zero, and all

parameters were computed at each iteration. We left the damping for the hypocenter parameters and station delays unchanged but increased the damping for the velocity parameters to 1.00 (Kissling et al., 1994) and to 10.00 (Husen et al., 2011) in two separate computations. Increasing only the damping of the velocity parameters in the final run prevents large perturbations in the velocities, especially in the poorly sampled layers, but allows for larger ones in the hypocenter parameters and station delays. This, together with the computation of all parameters at each iteration, leads to only fine

adjustments close to the previous solution (Husen et al., 2011) and the 1-D velocity model that minimize the total estimated location errors (Kissling et al., 1994).

The relative weighting factor between P and S readings must be set for the combined P and S inversion. This factor can account for the greater uncertainty in the S arrivals, which are always partially masked by the coda of P-waves. If we were

able to accurately account for this additional uncertainty in the S readings, depending on how it is assigned in the picking, we would set this parameter to 1.0. In practice, we cannot accurately account for this, but we can observe the effects of this phenomenon by looking at the number of readings by uncertainty class for each phase type and finding that higher uncertainties are generally assigned to the S phases. Given the relatively higher uncertainty of the S readings compared to the P readings in our dataset (Table S1), we expect this value to be relatively close to 1.0. Therefore, we used values of 1.0, 0.75,

and 0.5 for this parameter. For a value of 1.0, both phase types are equally weighted, while for a value of 0.5, the S readings are downweighted by half compared to the P readings. Moreover, the weighted RMS residuals of the S inversion are always higher than those of the P inversion. The main reason for this is the lower velocity of the S-waves, which means that the travel times and hence the absolute values of the residuals are larger, and the S-waves are more sensitive to smaller variations in velocity. However, this can also be seen as an indication of greater uncertainty in the S readings, which we were

unable to account for when picking the first arrival times.

The iterative process in the VELEST code is stopped, when the RMS residual or data variance ceases to decrease, or when the predefined number of iterations is reached. Due to lower damping values, different iteration types, and poor sampling in one of the layers, it is also possible that the inversion becomes unstable and must be stopped prematurely. For this reason,

and because some initial models are closer to the final solution than others, the total number of iterations was set between 2 and 10 for the first run. For the second run, the total number of iterations was set to 3, 5, 7, and 9. This also allowed us to examine and test the results of the inversions that diverged during the latter iteration steps. In general, the inversions with the lowest number of iterations were found to be unstable in the tests described below. We did not allow for low-velocity layers in the inversion as this resulted in larger instabilities that produced unrealistic and physically impossible velocity variations
in output models.

## 5.3 Tests

We performed several tests on the computed 1-D velocity models to check for any biases and to evaluate the stability of the solutions. The obtained hypocenter locations were systematically shifted by 10 km to greater depths and pseudorandomly shifted by 10 to 15 km in arbitrary directions before being introduced into another inversion run. The damping parameters of
this inversion run were identical to those of the second run, but this time we used the input station delays and computed the velocity model at every other iteration out of a total of nine, as suggested by Kissling (1995). If the velocities, station delays, and origin times remained relatively unchanged after this test and the hypocenters were relocated back to their initial positions, a stable solution was obtained and there should be no significant bias in the velocity model that could result from a systematic shift of the hypocenters. Because S-only inversion is more sensitive to the shift in depth, the systematic shift test
for this type of inversion was performed by shifting hypocenter locations by 7.5 km to greater depths. In addition, we tested the stability of each velocity model with the so-called high/low velocity test (Haslinger et al., 1999) by varying the P- and S-wave velocities of the obtained models by $\pm 0.5$ and $\pm 0.3$ km s$^{-1}$, respectively, and using them as initial models in an inversion run similar to that of the hypocenter shift test, but performing a simultaneous hypocenter-velocity inversion at each iteration. The models with large RMS residuals and large deviations in the velocity model and hypocenter parameters after
these tests were not considered suitable candidates for the minimum 1-D velocity model. The models that had the lowest RMS residuals performed well in the tests, implying that RMS residual can be used as an initial quality indicator for a 1-D velocity model. The tests can also be used to evaluate the coupling between the hypocenter and the velocity model parameters.

To see which velocity models and station delays yield reasonable hypocenter locations, we again used the VELEST code to relocate all well-locatable earthquakes (maximum azimuthal gap of 180° and a minimum number of first arrivals with an uncertainty class of less than 3) with each 1-D velocity model obtained in the inversion. The quality of each solution was also evaluated by relocating mining (quarry) blasts with known location and comparing the calculated station delays with the current knowledge of the geological structure in the region. Together with the hypocenter shift tests, relocation of quarry
blasts can provide an approximate estimate of the absolute uncertainty of the hypocenter location (Haslinger et al., 1999). Because hypocenter locations can be systematically shifted toward the surface due to an inappropriate velocity model, relocation of quarry blasts can give the false impression of a very good depth estimate. Therefore, the performance of an

individual model should not be judged solely based on the relocation of quarry blasts. By observing the consistent patterns of relocated quarry blasts among different models, one can also evaluate the performance of a seismic network for locating earthquakes in different parts of a study area.

## 6 Results

Throughout the modelling process, many 1-D velocity models were obtained from various initial 1-D velocity models and inversion parameters. We note that in the P-only inversion, the velocity models obtained from the initial models with low and high velocities only partially converged towards the solution obtained with the three initial models derived from the independent studies (R1D; Brückl et al., 2007; Šumanovac et al., 2009) and performed comparatively poorly in the tests. This suggests that it is very difficult to obtain a stable solution when the starting point in the parameter space is relatively far from the true model. For this reason, we used the two best performing models computed in the P-only inversion (MP1, MP2) as initial models for the S-only inversion by multiplying the velocity values by a constant $v_P/v_S$ value of 1.73. The performance of the computed models was evaluated using the RMS residuals, stability tests, and relocations. We also selected the two best performing models of the S-only inversion (MS1, MS2) and used them, together with the MP1 and MP2 models, as initial models for the combined P and S inversion. For all inversions, the damping value for the velocity parameters was set to 10.0 in the second run, as we obtained better convergence and slightly lower RMS residuals. In practice, however, no large difference was observed in well-sampled layers when the value was set to 1.0. Since the velocity of the surface layer in the S-only inversion rapidly dropped to unrealistically low values, we individually damped the velocity in this layer by setting the damping to 10.0 and 50.0. We then decided to use the former value, which still allowed some velocity perturbations in the surface layer. This damping value for the S velocity of the surface layer was also kept in the initial models for the combined P and S inversion. We constructed four initial P and S models. Two where we used the same damping value (10.0) also for the P velocity of the surface layer and the other two where the P velocity of the surface layer was left undamped. Since the P- and S-only inversions served as intermediate steps for the combined P and S inversion, we only discuss the results of the latter. Nonetheless, the results of P- and S-only inversions serve as another indicator of how well the velocity is constrained in each layer. The best performing models of P- (MP1, MP2) and S-only (MS1, MS2) inversions are shown in Fig. S2, along with the final models of the combined P and S inversion and their median velocities.

Using the four initial P and S velocity models, 80 velocity models (final models; Fig. S2) were obtained with different total number of iterations in the combined P and S inversion. The P velocities of the final models show very good convergence and are well constrained for layers between 8 and 26 km depth, with a maximum difference between the 15th and 85th percentiles (hereafter referred to as the percentile difference) of 0.10 km s$^{-1}$ (Figs. 5 and S2, Table S3). Layers between 0 and 8 km depth show worse convergence, with the percentile differences between 0.12 and 0.15 km s$^{-1}$. At greater depths, between 26 and 34 km, the P velocities diverge the most, with the percentile differences of 0.18 and 0.20 km s$^{-1}$. The S

velocities are well constrained for layers between 4 and 26 km depth, with the percentile differences of less than 0.09 km s[-1] (Figs. 5 and S2, Table S4). In layers between 0 and 2 km and between 30 and 34 km depth the S velocities converged to a similar extent, with the percentile differences ranging from 0.10 to 0.12 km s[-1]. Despite some relatively small values of the percentile differences, we estimate that the P and S velocities are poorly constrained at depths below 34 km because only 9-618 rays sampled the corresponding layers. The final RMS residuals for the combined P and S inversion dataset are mostly

(85th percentile) below 0.323 s, with the lowest value of about 0.288 s (Fig. S5), and generally show a reduction of up to 13-22 percent compared to the first iteration. Based on the RMS residuals, stability tests, and relocations, we selected a (minimum) 1-D P and S velocity model for which we show the detailed results.

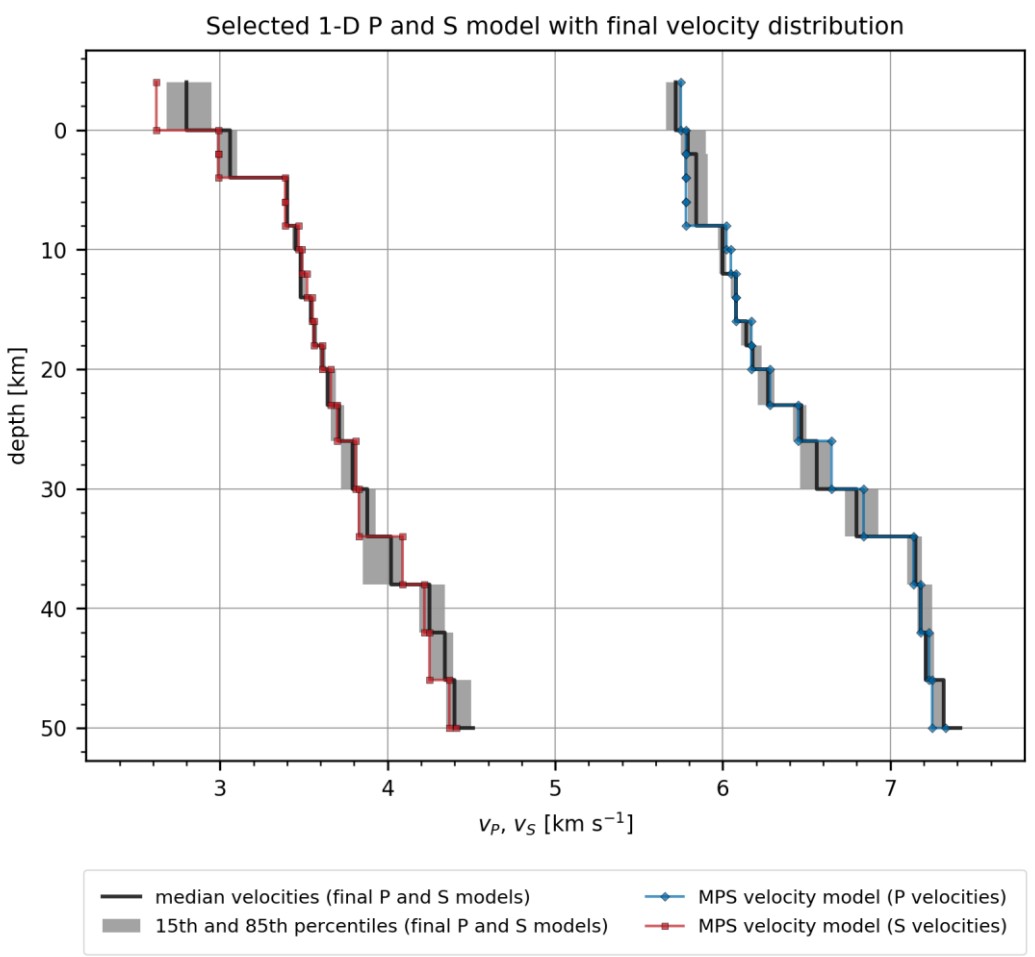

**Figure 5: Selected P and S velocity model (blue) chosen after examining the results of the stability tests, relocations, and final RMS residual values. Also shown are the median velocities (black line) and velocity percentiles (grey area) of the final P and S models as calculated in each layer. Corresponding values are given in Tables S3 and S4.**

### 6.1 Minimum P and S velocity model (MPS)

The minimum P and S velocity model (MPS) shown in Fig. 5 was computed from the MP1 and MS1 models with eight total iterations in the first run and seven total iterations in the second run. A damping value of 10.0 was used for the P and S velocities of the surface layer. The model shows constant P velocity of 5.78 km s$^{-1}$ between 0 and 8 km depth. At 8 km depth, we observe a jump in P velocity to 6.02 km s$^{-1}$ and then a gradual increase to 6.28 km s$^{-1}$ at 20 km depth. In the deeper layers, P velocity starts to increase more rapidly, from 6.45 km s$^{-1}$ at 23 km depth to 7.14 km s$^{-1}$ at 34 km depth, where the largest jump is observed. Conversely, the S velocities show a very significant jump at 4 km depth from 2.99 km s$^{-1}$ to 3.39 km s$^{-1}$ and then a gradual increase to 3.83 km s$^{-1}$ at 30 km depth. Another more rapid increase in S velocity is observed at 34 km depth, where it increases to 4.09 km s$^{-1}$. This jump in S velocity occurs in a layer extending to 38 km and coincides with the largest jump in the P velocities. Moreover, both the P and S velocities of the MPS model strongly resemble the values and features of the median velocity model. Fewer than 1,000 rays (P and S readings combined) penetrated the layers between 30 and 38 km depth, which were sampled in only a few directions because earthquakes in the study area occur only at shallower depths (Fig. 6). This prevents us from better constraining the velocities in these layers. Layers below 38 km were sampled with 168 rays or less, which means that velocities at these depths remained unconstrained.

The P-wave station delays show the same general trend for all final models and were referenced to the seismic station with many high-quality picks and a location approximately in the middle of the selected seismic stations (Fig. 7). Seismic stations in the west show large negative delays that gradually transition to positive delays in the east and south. We observe relatively large positive station delays in the Krško basin (KB), the Sava basin (SB), and other sedimentary basins such as the Barje and Gorenjska basins (BB and GB). Slightly negative station delays were computed for the region of magmatic and metamorphic rocks in the Eastern Alps (north-east of the Labot fault – LF). West of this region, positive station delays extend along the Periadriatic fault (PAF). In the southern part of the study area, positive station delays extend approximately in the direction of NW-SE along the Adriatic coast, starting north of the Rijeka region (RR). These positive delays are less pronounced at the stations located more inland, except for the southernmost seismic station (UDBI), which shows larger positive station delay. Deeper and large-scale velocity variations in the crust are more strongly reflected in the delays of seismic stations with limited azimuthal coverage at the periphery of the study area. This is mainly due to the longer ray paths, which pass through the poorly sampled parts of the crust away from the volume sampled by most rays (Fig. 6). The S-wave station delays are also computed in the combined P and S inversion but are referenced to the P-wave station delays and much harder to interpret, which is why we do not discuss them here.

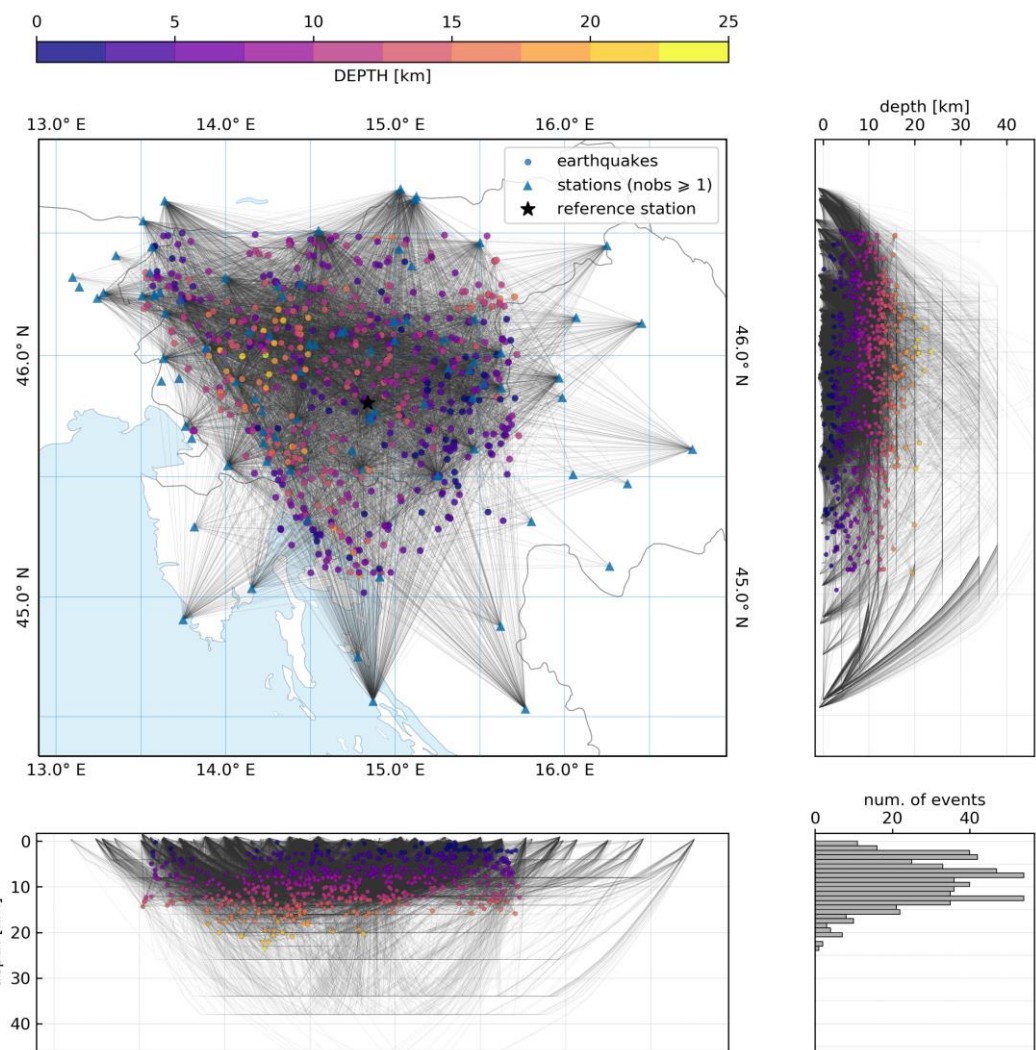

**Figure 6: Seismic rays and hypocenters computed for the MPS velocity model. The right and bottom panels show the hypocenters of earthquakes and rays projected on N-S and W-E oriented profiles, respectively. The histogram in the lower right corner shows the number of earthquakes in 1 km depth bins. The seismic stations used in the inversion are also shown.**

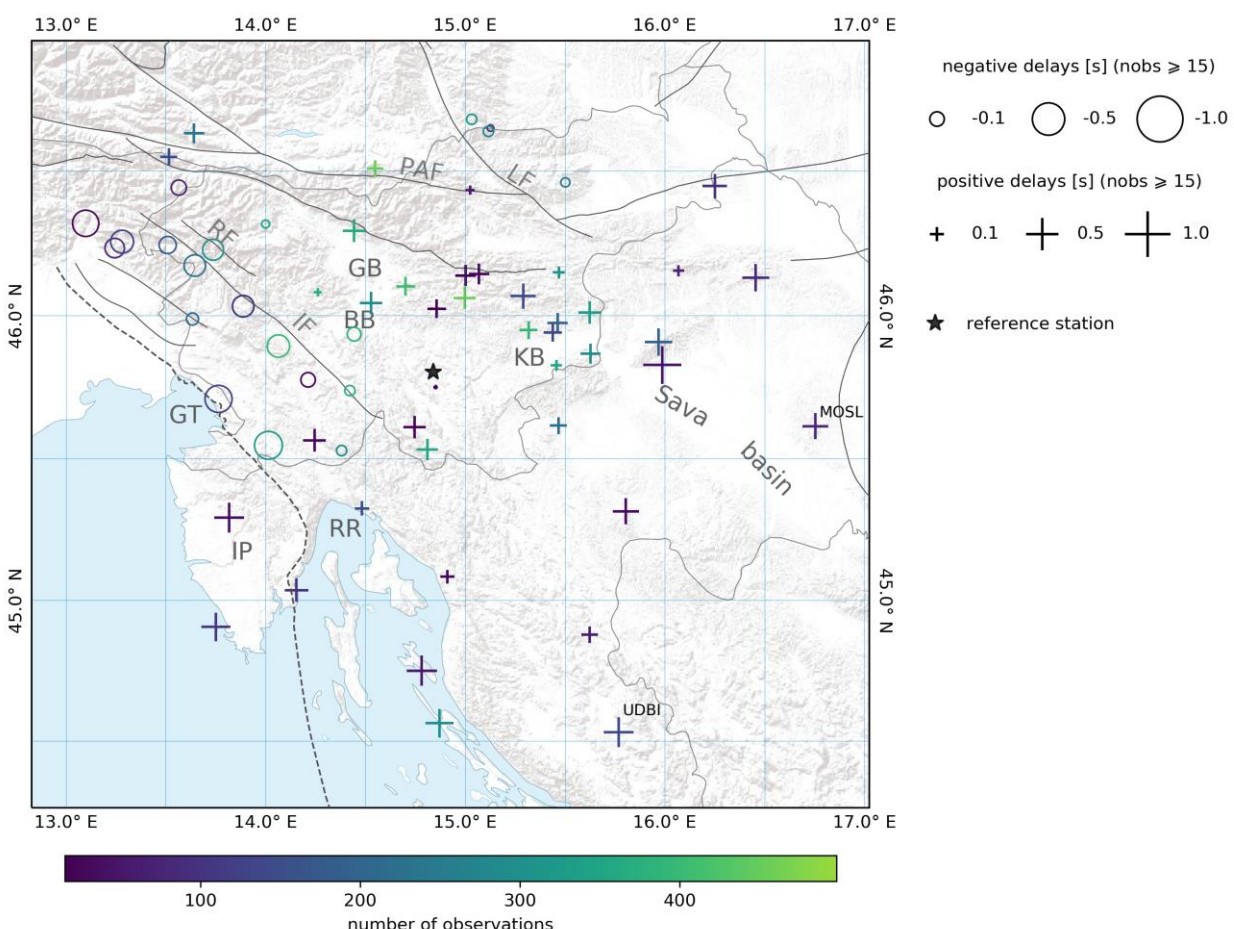

**Figure 7: P-wave station delays computed for the MPS velocity model. Station delays are shown only for stations with at least 15 observations. Black star marks the reference station (see main text for details). Refer to Fig. 1 for the list of geographical names and faults. Shaded relief is shown in the background (Esri, USGS, NOAA).**

The selected model performed better than other models in the stability tests. In the high/low velocity test, we varied the P

and S velocities of the MPS model by ±0.5 and ±0.3 km s$^{-1}$, respectively, and performed another inversion run. The velocities of the models obtained in the high/low velocity test converged close to the velocities of the MPS model (Fig. 8 and Table 1) and indicate that a stable solution was obtained for layers between 0 and 38 km depth. In this depth range, the P velocities of the layers between 2 and 8 km and between 20 and 26 km depth deviated the most from the values of the MPS model after the high/low velocity test was performed. In the low velocity test, the P velocity of the layer between 23 and 26

440 km depth converged to 0.15 km s$^{-1}$ of its value in the MPS model and showed the largest deviation among the recovered P-wave velocities. Nevertheless, it fully converged in the high velocity test. The only P velocity layer that did not fully converge in both the high and low velocity tests begins at 20 km depth. Since the output P velocity of this layer was lower in the high and low velocity tests, the velocity computed in the MPS model could be too high by at most 0.10-0.12 km s$^{-1}$. This

layer also marks the transition from 2 km to 3 km thick layers, which could lead to some instabilities in the high/low velocity

test. In addition, a significantly different number of rays refracted between 20 and 23 km depth for the high/low velocity test

and the combined P and S inversion. In the high velocity test, many rays refracted already in the layer between 2 and 4 km

depth, while in the combined P and S inversion and the low velocity test, these rays started to refract in the layer between 4

and 6 km depth. This could explain the 0.12 km s$^{-1}$ difference in P velocity that we see for layers between 2 and 8 km depth

after the high velocity test. Such a change in geometry of the rays, and hence the velocity of some layers, is to be expected

due to the large velocity variations we introduce to models in the beginning of the high/low velocity test. The P velocities

between 2 and 8 km depth increased in the last two iterations of the high velocity test inversion, despite having converged to

0.06 km s$^{-1}$ of their values in the MPS model already at the fifth iteration, where the data variance and RMS residual were

the lowest. The hypocenter parameters did not change significantly (Table 1). For the high/low velocity test only, we

undamped the velocities of the surface layer.

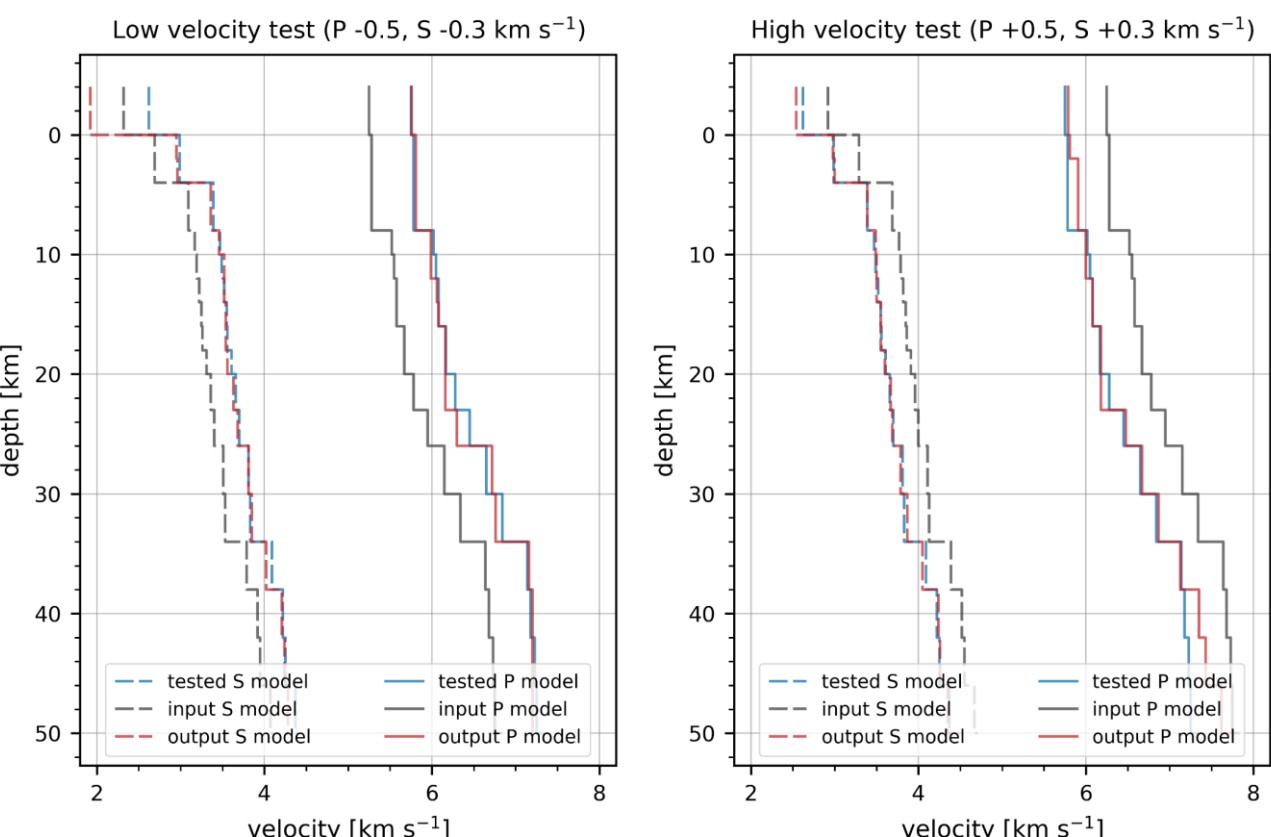

**Figure 8: High/low velocity test for the MPS velocity model (blue), where we varied the P and S velocities by ±0.5 and ±0.3 km s$^{-1}$, respectively and used them as the input velocity model (grey) for another inversion run with 9 iterations. The output of this test is shown in red.**

Table 1: The results of the high/low velocity test for the MPS velocity model, given as average and standard deviation of differences between the values obtained after the inversions with the varied models and the final parameter values. The velocity values are calculated only for the well sampled layers between 0 and 38 km. The statistics for the epicenter and hypocenter values were calculated from the lengths of vector differences.

| Input velocity variation [km s$^{-1}$] | Epicenter [km] | Hypocenter [km] | Origin time [s] | P velocity [km s$^{-1}$] | S velocity [km s$^{-1}$] | P station delays [s] | S station delays [s] |
|---|---|---|---|---|---|---|---|
| -0.5 P, -0.3 S | 0.10 ± 0.14 | 0.59 ± 0.54 | -0.03 ± 0.03 | -0.02 ± 0.06 | -0.02 ± 0.02 | 0.00 ± 0.01 | 0.09 ± 0.11 |
| +0.5 P, +0.3 S | 0.08 ± 0.12 | 0.28 ± 0.43 | 0.04 ± 0.03 | 0.02 ± 0.06 | -0.00 ± 0.02 | 0.01 ± 0.01 | 0.02 ± 0.04 |

After performing the systematic and pseudorandom hypocenter shift tests, the hypocenters were relocated close to the locations determined for the MPS model (Figs. 9 and 10), while the velocity model parameters mostly showed only small deviations from the values of the MPS model (Table 2). Systematic changes in the origin times are observed for the pseudorandom hypocenter shift test. The resulting positive shift in origin times for the selected model is one of the smallest among the final models, and we consistently observe such a shift in origin times for all models after performing the pseudorandom hypocenter shift tests. The reason for this systematic shift could be in the pseudorandom hypocenter shift test itself. Since we did not allow the earthquakes to be shifted above 0 km depth, more earthquakes were shifted to the greater depths. This systematically increased the travel times of at least the direct rays, which represent most (67 percent) of the rays computed for the MPS model. The increased travel times were then compensated in the inversion of the pseudorandom hypocenter shift test by an increase in P velocity of 0.10-0.11 km s$^{-1}$ for layers between 2 and 8 km depth. This increase in velocity was reflected in the positive systematic shift of the origin times, as the hypocenters were eventually relocated close to their original positions and were even 270 m shallower on average. The velocity increase also resulted in a small velocity jump at 2 km, which caused some direct rays from the shallow events to be modelled as refracted rays or existing refracted rays to be refracted earlier, effectively shortening travel times. This test suggests that the P velocities in the layers between 2 and 8 km depth show some coupling with the hypocenter parameters, which is consistent with the observations made in the high velocity test (Fig. 8) and worse P velocity convergence of the final P and S models (Fig. 5) in this depth range. From this we can safely conclude that the P velocities are relatively less well constrained in these layers, but we cannot say whether the absolute velocities of the MPS model are too high or too low relative to the actual velocities.

The systematic shift test significantly affected only the S velocities of the layers between 0 and 4 km depth, which decreased by 0.17 km s$^{-1}$ and already showed worse S velocity convergence of the final P and S models than the other layers above 26 km depth. Compared to the changes in the velocity models after the pseudorandom shift test, this implies that in the less well constrained layers, the S velocities (and S station delays) are more sensitive to systematic shifts in depth, while the P velocities, and thus the origin times, are more likely to change when the hypocenters are shifted pseudorandomly. For the well constrained layers the differences in velocities between models resulting from the shift tests and the MPS model range

between -0.02 and 0.07 km s$^{-1}$ and we observe very little coupling between the velocity and the hypocenter parameters. This means that a variation in the hypocenter or velocity parameters was not compensated by a large variation in the velocity or hypocenter parameters, respectively.

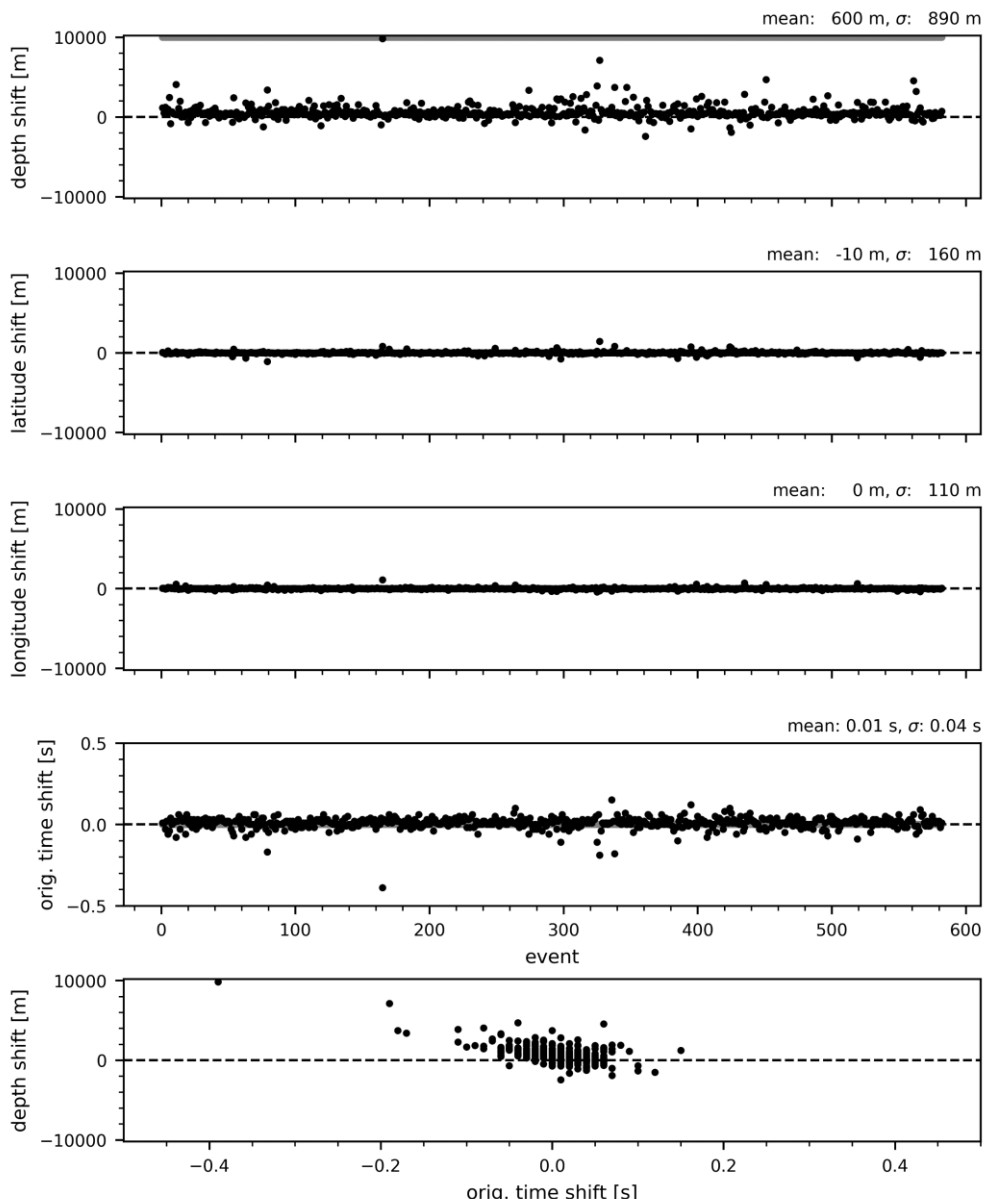

**Figure 9: Systematic hypocenter shift test. Hypocenters obtained with the MPS velocity model were shifted systematically in depth by 10 km (grey dots at the top of the first plot) and used as an input in another inversion run with nine iterations. Black dots show the resulting shifts in the hypocenter parameters remaining after this test. All shifts are referenced to hypocenter parameters obtained with the MPS velocity model.**

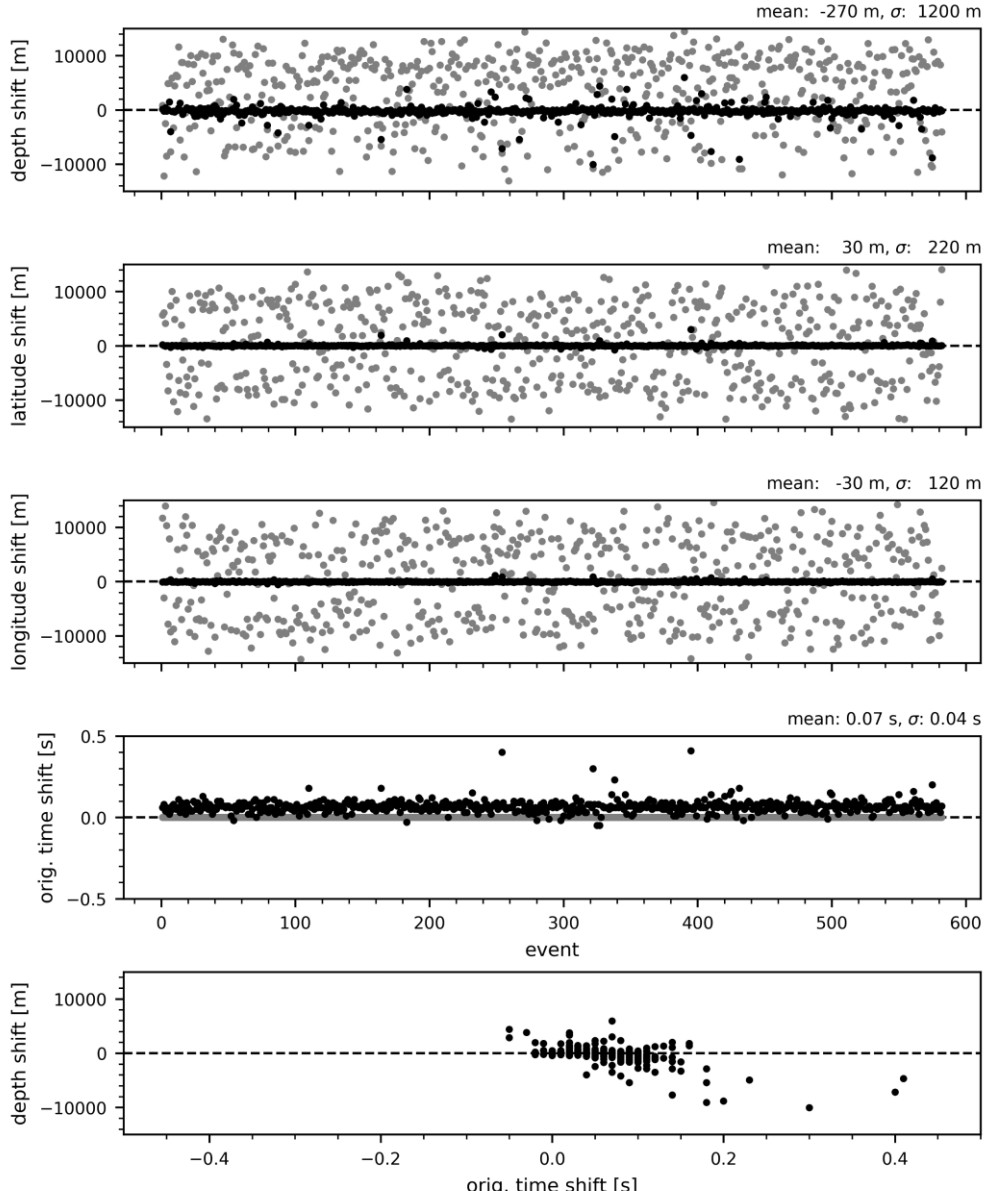

Figure 10: Pseudorandom hypocenter shift test. Hypocenters obtained with the MPS velocity model were shifted along pseudorandomly generated vectors by 10 to 15 km (grey dots in the first three plots) and used as an input in another inversion run with nine iterations. Black dots show the resulting shifts in the hypocenter parameters remaining after this test. All shifts are referenced to hypocenter parameters obtained with the MPS velocity model.

**Table 2: The results of the systematic and pseudorandom hypocenter shift tests for the MPS velocity model, given as average and standard deviation of differences between the values obtained after another inversion and the final parameter values. The velocity values are calculated only for the well sampled layers between 0 and 38 km. The statistics for the epicenter and hypocenter values were calculated from the lengths of vector differences.**

| Input hypocenter shift [km] | Epicenter [km] | Hypocenter [km] | Origin time [s] | P velocity [km s$^{-1}$] | S velocity [km s$^{-1}$] | P station delays [s] | S station delays [s] |
|---|---|---|---|---|---|---|---|
| 10 (Z) | 0.12 ± 0.16 | 0.73 ± 0.81 | 0.01 ± 0.04 | -0.02 ± 0.04 | -0.05 ± 0.05 | 0.01 ± 0.02 | 0.17 ± 0.06 |
| 10-15 (XYZ) | 0.13 ± 0.13 | 0.62 ± 1.09 | 0.07 ± 0.04 | 0.01 ± 0.05 | -0.02 ± 0.03 | 0.01 ± 0.02 | 0.03 ± 0.05 |

The RMS residual obtained for the MPS model after the inversion was 0.296 s. We selected 3,282 earthquakes with a maximum azimuthal gap of 180° and at least 10 first P and 5 first S arrivals with an uncertainty class of less than 3. By relocating them with the velocities and station delays of the MPS model (Fig. 11), we obtained the RMS residual of 0.323 s. Compared to the relocation of the same earthquakes with the R1D model and using the same observations, the reduction in the RMS residual is about 22 percent. This reduction is also clearly visible when looking at the distribution of the residuals (Fig. S6). A look at the distribution of seismicity shows that 75 earthquakes, or about 2 percent, were relocated to depths between 0 and 1 km. There are several possible explanations for this. These include possibly overlooked quarry blasts in the dataset and poor geometry of the stations used to locate these earthquakes, most of which occurred at the periphery of our study area (Fig. S7). This also implies that the resolved velocity structure could bias the depth of earthquakes in the poorly sampled regions. The gap in seismicity above a depth of about 7 km in western Slovenia (around 14.0° E longitude) can be observed in both the relocations and the routinely located seismicity. Conversely, earthquakes in the eastern part of the study area occur at shallower depths and are absent in some parts already below about 10 km depth.

The relocation of quarry blasts was performed for 92 events with a maximum azimuthal gap of 180° and at least 8 first P and 5 first S arrivals with an uncertainty class of 2 or better (Fig. S8). The average mislocations from known locations are 1.22 and 2.15 km for epicenters and hypocenters, respectively. We also relocated the same quarry blasts using the R1D model and obtained significantly larger average mislocations of 1.53 km for epicenters and 7.37 km for hypocenters, suggesting that this model systematically shifts hypocenters to greater depths (Fig. S9). For all final models, we observe consistent mislocation of hypocenters to greater depths (below 5 km) for two blasts in south-western and northern Slovenia. The blast in northern Slovenia and another one at the south-eastern Slovenian border also show a large mislocation in their epicenters. The readings for these blasts need to be validated.

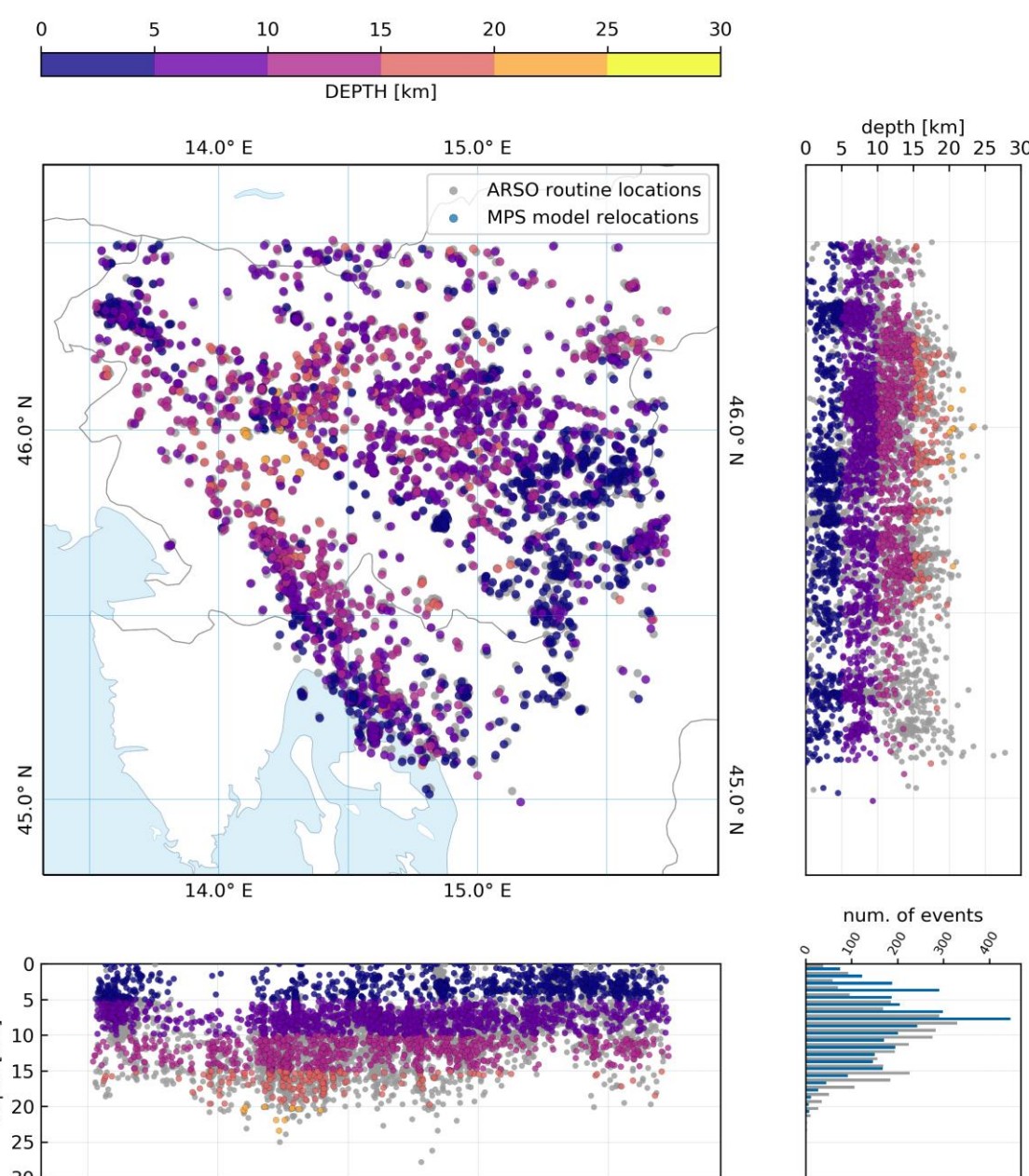

**Figure 11: Relocation of 3,282 earthquakes with the MPS velocity model (velocities and station delays). The relocation included earthquakes between 2004 and 2018 with a maximum azimuthal gap of 180° and at least 10 P and 5 S arrivals with an uncertainty class below 3. The right and bottom panels show the hypocenters of earthquakes projected on N-S and W-E oriented profiles, respectively. The histogram in the lower right corner shows the number of earthquakes in 1 km depth bins for routine locations (grey) and relocations (blue).**

**6.2 Regional subdivision into three subregions**

To gain a better insight into the velocity structure of the crust, we divided the study area into three subregions (Fig. 12),
which were defined mainly based on the station delays (Fig. 7) and the distribution of the relocated seismicity (Fig. 11). We
reselected the seismic stations and earthquakes that were relocated using the MPS model and performed the inversion for
each subregion separately. To include more earthquakes in the selection procedure, we decreased the minimum vertical
distance between earthquakes within a single cell from 2 to 1 km and reduced the cell size to 5 km. For the south-western

(SW) subregion, we also reduced the number of readings per earthquake to include at least 8 first P and 5 first S arrivals. The
results of the reselection procedure are shown in Table 3. The inversion procedure was the same as for the regional
inversion, but this time we used the MPS model as the only initial model for the inversion. Since we used the regional model
directly as the initial model for the combined P and S inversion and the reference stations changed, we kept the two-run
approach for the inversion. For each subregion we also performed the stability tests and selected the best performing model.

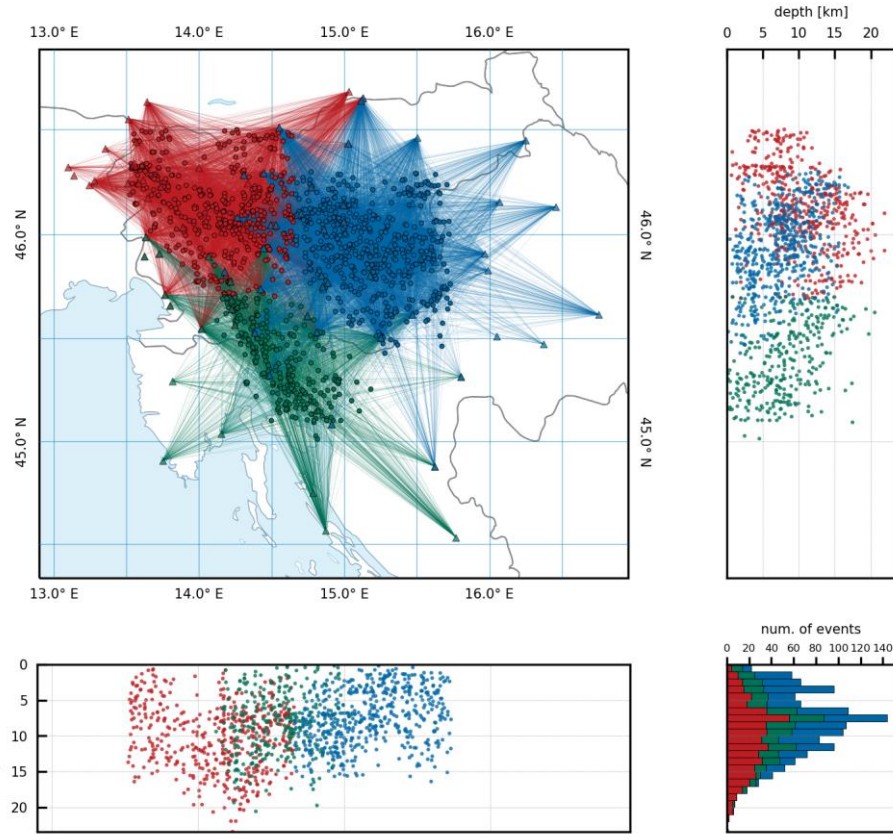

**Figure 12: Division of the study area into the eastern (E; blue), north-western (NW; red), and south-western (SW; green)
subregions. Great-circle rays connecting earthquake station pairs are shown. The right and bottom panels show the hypocenters of
earthquakes projected on N-S and W-E oriented profiles, respectively. The histogram in the lower right corner shows the number
of earthquakes in 1 km depth bins for each subregion.**

The velocities of the layers below 23 km depth are poorly constrained for all subregions because less than 100 rays penetrate deeper due to the short epicentral distances, the relatively shallow seismicity in the region, and large deviations in the high/low velocity tests. The high/low velocity tests show that a stable solution was obtained for layers between 0 and 20 km depth in the E subregion and between 0 and 23 km depth in the NW subregion. The models computed for the SW region showed relatively better performance for layers between 0 and 23 km depth but performed significantly worse in the stability tests compared to the other two subregions. This was to be expected due to a fewer number of readings used for the inversion and inferior earthquake-station geometry. Thus, we have to be careful when interpreting the results of the SW subregion. Considering the ray distribution and the high/low velocity tests, we focus only on the velocities between 0 and 23 km depth. For all subregions, we see a reduction in the RMS residuals of about 21-30 percent compared to the regional model (Table 3).

**Table 3: Results of the selection procedure and the lowest RMS residuals of the final P and S models obtained for each subregion.**

| Region | Number of earthquakes | Number of first P arrivals | Number of first S arrivals | RMS residual of the selected model [s] |
|---|---|---|---|---|
| Eastern (E) | 528 | 9,937 | 7,408 | 0.235 |
| North-western (NW) | 481 | 8,876 | 7,682 | 0.228 |
| South-western (SW) | 300 | 3,340 | 3,301 | 0.207 |
| Regional (MPS) | 582 | 13,034 | 10,134 | 0.296 |

By comparing the P velocities of the selected models among the subregions (Fig. 13, Tables S10, S11, and S12), we immediately notice lower velocities of the E subregion in layers above 8 km depth. Compared to the regional (MPS) model the P velocities for these layers are lower above 4 km depth for the E subregion and higher for the other two subregions. All subregions show very similar P velocities between 8 and 23 km depth, which are also close to the regional velocities. In this depth range, the main difference among them is found between 12 and 18 km depth, where the SW subregion shows 0.04-0.12 km s$^{-1}$ higher P velocities. Below 23 km depth, the P velocities start to diverge significantly among the subregions. Apart from the jump in P velocity at 8 km depth for the E subregion, we do not observe other significant jumps in velocity, but rather a gradual increase in P velocity for the well-resolved layers. Conversely, the S velocities of the selected models (Fig. 14, Tables S10, S11, and S12) are very similar in the layers above 6 km depth and diverge significantly below this depth. The E subregion shows the lowest S velocity for the layer starting at 0 km depth and consistently higher S velocities for all other well-resolved layers also if compared to the regional velocities. The S velocities of the NW and SE subregions are very similar, except for the layers between 4 and 8 km and between 16 and 20 km, where the NW region shows higher velocities. Compared to the regional model, the S velocities of the models for the NW and SE subregion are lower. We

observe a jump in velocity at 4 km depth in all subregional models, which is consistent with the regional model. The model for the E subregion also shows a less pronounced but still significant velocity jump at 2 km depth.

We also calculated $v_P/v_S$ values from the P and S velocities of the MPS model and the subregional models (Fig. 15, Tables S10, S11, and S12). All models consistently show high $v_P/v_S$ values in the upper 4 km depth. The $v_P/v_S$ values between 4 and 16 km depth in the E subregion are low but close to 1.73 and drop to about 1.65 in the depth range between 16 and 23 km. In the NW and SW subregions, $v_P/v_S$ is consistently above 1.73 between 4 and 16 km depth and close to this value in the depth

range between 16 and 23 km depth. Below 4 km depth, $v_P/v_S$ of the regional model is consistently close to 1.73 and shows no significant deviations.

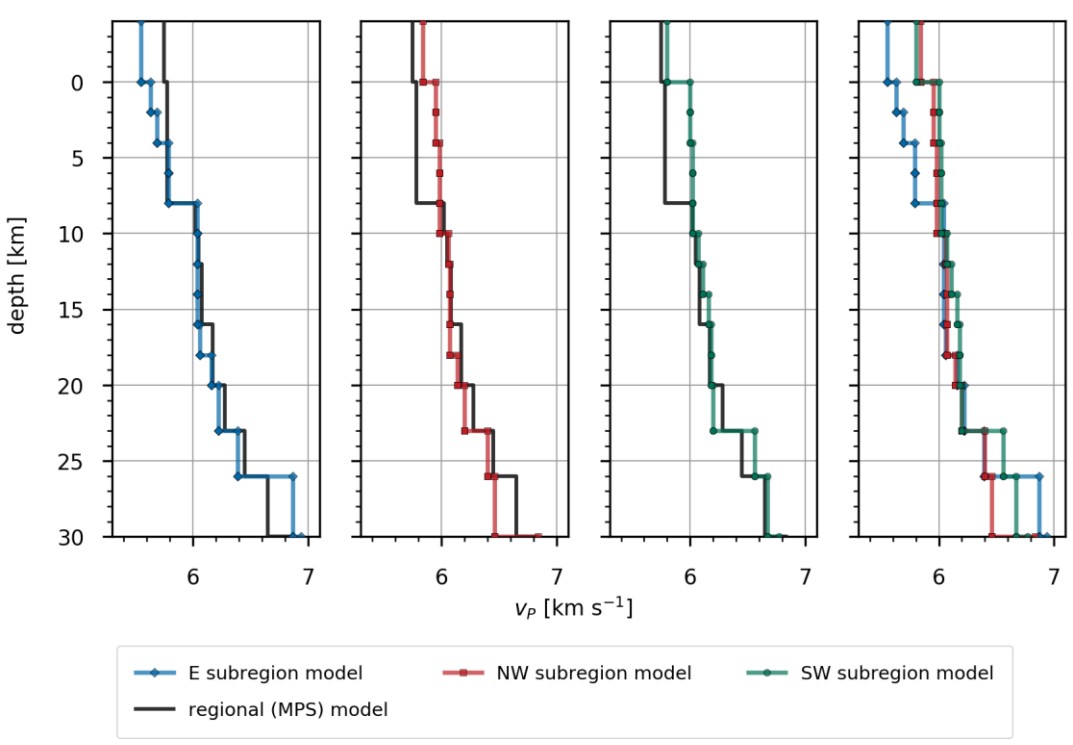

**Figure 13: P velocities computed for a particular subregion. Thick black line shows P velocities of the regional (MPS) model.**
**Corresponding values are given in Tables S10, S11, and S12.**

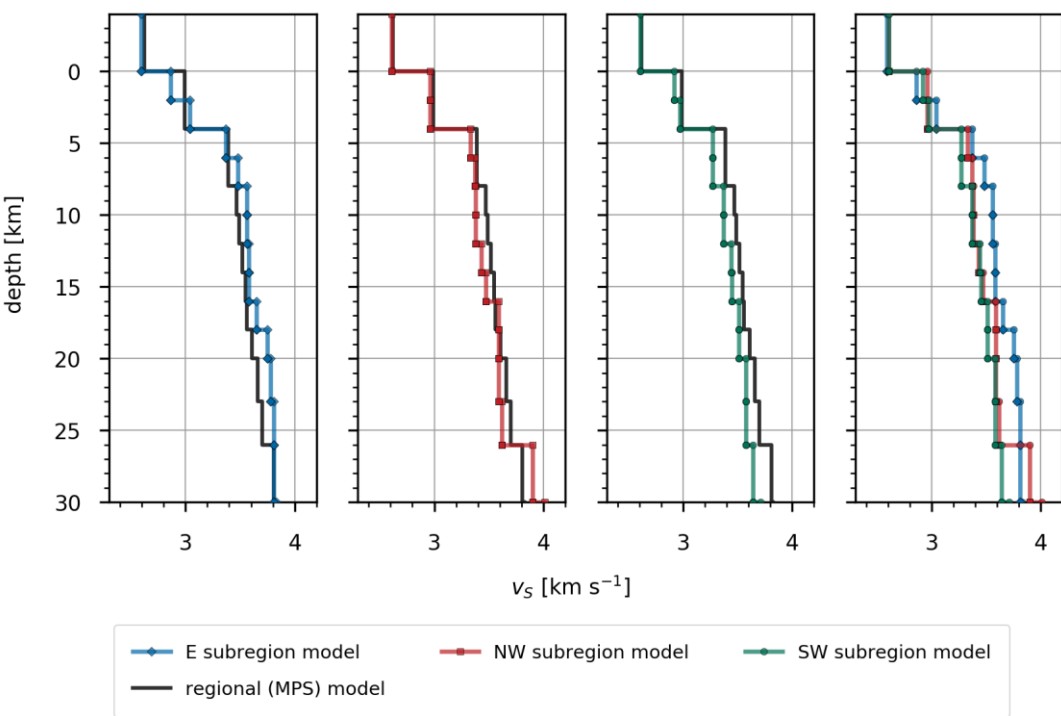

**Figure 14: S velocities computed for a particular subregion. Thick black line shows S velocities of the regional (MPS) model. Corresponding values are given in Tables S10, S11, and S12.**

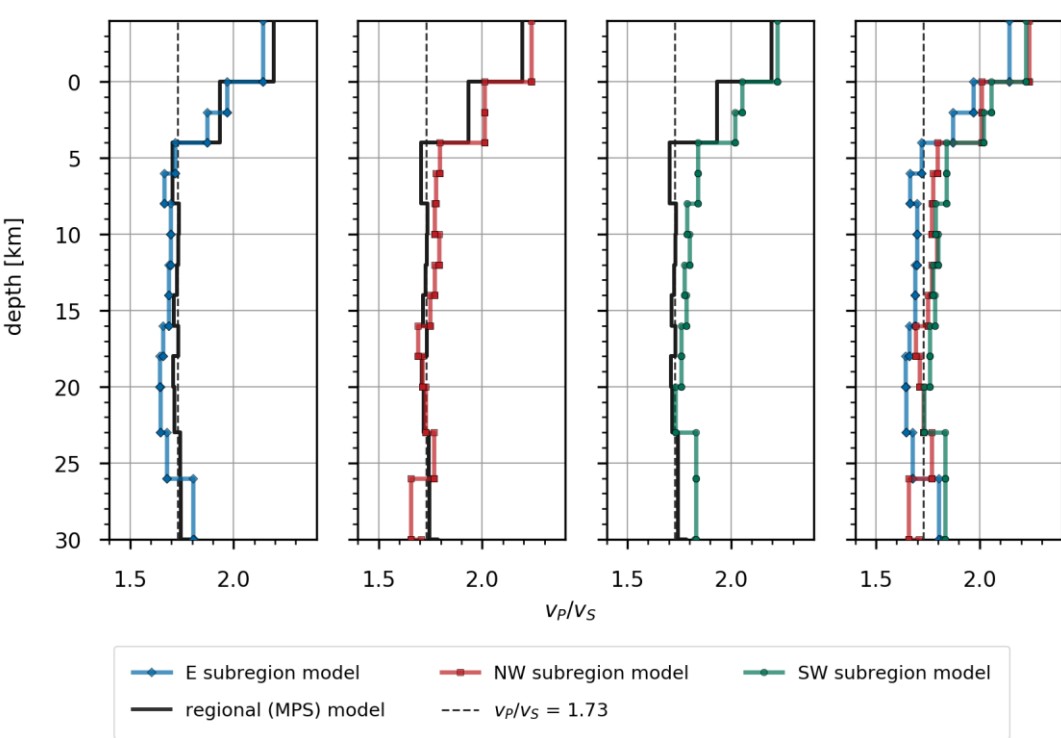

**Figure 15: P to S velocity ratio ($v_P/v_S$) values calculated for a particular subregion. Thick black line shows $v_P/v_S$ values of the regional (MPS) model. Corresponding values are given in Tables S10, S11, and S12.**

## 7 Discussion and conclusions

One of the goals of this study was to complement the results of previous studies and to expand our knowledge of crustal structure in the region. The seismic ray distribution (Fig. 6) shows that the upper crust is adequately sampled. This was made possible by the modernization of the SNRS (Vidrih et al., 2006; Jesenko & Živčić, 2018) and the deployment of additional seismic stations in Croatia within the VELEBIT and AlpArray projects (Molinari et al., 2016). Considering the results of the stability tests and the convergence of the final regional models (Fig. 5) we estimate that a good solution was obtained for depths between 0 and 26 km. The fact that the layers below 26 km were sampled by a comparatively small number of subvertical rays (Fig. 6), ranging from 9 to 1,163 rays, limits the ability of the inversion to resolve the velocity structure of the lower crust in more detail. Nevertheless, the presence of at least 618 rays, the convergence of the final regional models, and the simple velocity structure suggest that at least the average velocity has been resolved for depths between 26 and 38 km.

Several features are observed in the computed regional velocity model. Rather prominent P velocity jumps appear at 8 km and below 23 km depth (Fig. 5). Large velocity jumps are observed in the lower crust at the interfaces between 23 and 34 km depth, where the P velocity starts to increase more rapidly. As expected, we do not observe a single sharp increase in velocity that would indicate a clear depth of the Moho discontinuity. Rather, the depth interval of the rapid increase in P velocity suggests a change in crustal structure or highly variable Moho topography. The apparent increase in velocity gradient with depth at interfaces between 23 and 30 km could therefore indicate the transition from upper to lower crust, in agreement with the results of Magrin & Rossi (2020) for the northern Adria, including the NW Dinarides. The transition was interpreted to occur at a P-wave velocity of about 6.4 km s$^{-1}$. The last and largest jump in P velocity at 34 km depth coincides with the only prominent jump in S velocities below 4 km depth. This is the only feature of the MPS model that could indicate an average Moho depth in the study area and would place it roughly between 34 and 38 km, consistent with previous studies (e.g., Stipčević et al., 2020). The velocity jumps are unlikely to be as pronounced as in reality, as the velocity in each layer approaches the average of the 3-D velocity variations sampled by the rays. Large lateral variations may therefore mask large vertical velocity discontinuities, meaning that some of them were probably not resolved by this method.

P velocities in the E subregion between 0 and 8 km depth are much lower compared to the other two subregions (Fig. 13, Tables S10, S11, and S12) and are related to the presence of deep sedimentary basins at the periphery of the Pannonian basin such as the Krško basin. The reason for the higher P velocities in the NW and SW subregions is most likely the thick cover of carbonate rocks. In the depth range from 8 to 23 km, the P velocities are very similar in all subregions and close to the regional model. The main difference among the subregions in this part of the crust is found between 12 and 18 km, where the SW subregion shows 0.04-0.12 km s$^{-1}$ higher P velocities. The S velocities of the selected subregional models (Fig. 14, Tables S10, S11, and S12) are very similar in the layers above 6 km depth and diverge considerably below this depth. The E

subregion shows the lowest S velocity for the layer starting at 0 km depth and consistently higher S velocities for all other well-resolved layers. The higher S velocities in the E subregion indicate a more compact material, while the lower S velocities in the NW and SW subregions can probably be associated with highly fractured rocks. Furthermore, this also suggests denser volcanic crust in the E subregion, the origin of which can be attributed to the rifting in the Pannonian basin (e.g., Fodor et al., 1998), and less dense thickened crust of continental origin in the NW and SW subregions that formed as a result of the underthrusting of the Adria (e.g., Tari, 2002; Brückl et al., 2010). The largest jump in S velocity occurs at 4 km depth in all subregional models and is also consistent with the regional model. This shallow jump in S velocity indicates a transition from the overlying layers being more saturated with fluids than the underlying layers. A less pronounced, but still significant, jump in S velocity that is apparent in the model for the E subregion at 2 km depth likely indicates a transition from loose to more compacted sediments. In all subregional models, depth intervals of slow velocity increase are observed, ranging from 8 to 10 km. Such features could result from thick homogeneous layers where seismic velocity is mainly controlled by pressure and temperature gradient. The $v_P/v_S$ values calculated from the P and S velocities of the MPS model and the subregional models (Fig. 15, Tables S10, S11, and S12) are consistently high in the upper 4 km depth. Below this depth, $v_P/v_S$ values are relatively high until a depth of 16 km, where they begin to decrease for all subregions. Below 4 km depth, $v_P/v_S$ of the regional model is consistently close to 1.73 and shows no significant variation.

In terms of absolute P velocities, the obtained velocity models show significantly higher velocities above 30 km depth compared to the R1D model (Fig. 16). Only the P velocities of the layer between 2 and 4 km depth obtained for the E subregion are lower compared to the routine model. In comparison with the S velocities of the R1D model, the S velocities computed in this study are lower in the upper 4 km depth and mostly higher in the deeper layers (Fig. 17). We see large differences in velocities between our results and the R1D model, except when compared to the S velocities of the NW and SW subregions. The velocities computed for the subregions in the well-constrained layers between 0 and 23 km depth can also be compared with some other velocity models obtained in previous studies. As already observed by Michelini et al. (1998), we see that P velocities in the upper 6 km differ significantly between western and eastern Slovenia. We also obtained similar P velocity values in the first 6 km of the crust. The velocities of the deeper crust in the west, at 13 and 20 km, obtained by Michelini et al. (1998) also agree well with our results, but on the other hand we obtained higher P velocities for the E subregion. The P velocities in the deeper parts of the E subregion seem to be more in agreement with the velocity values determined by Kapuralić et al. (2019). The P velocities extracted from the 3-D model of Magrin & Rossi (2020) at the point near the reference station for our regional inversion (NAC2 14.86E 45.83N) agree well with the P velocities of the MPS model in the upper 34 km depth, with some relatively large differences in the depth intervals 4-8 km and 18-23 km. The S velocities show large differences in the upper 12 km depth and a very good agreement between 12 and 38 km depth. Compared to the velocities of Magrin & Rossi (2020) at another point of their model (NAC2 14.28E 46.05N), we obtained slightly lower P velocities for the NW subregion between 0 and 23 km depth, while the S velocities between 4 and 23 km depth are in very good agreement with their results. In the depth interval between 6 and 23 km depth, the P

velocities of the NW subregion are also lower than those obtained by Brückl et al. (2007; ALP01 460 and 520 km), which was expected since this model was also one of the models used for the construction of the NAC2 model (Magrin & Rossi, 2020). The model of Magrin & Rossi (2020, NAC2 14.35E 45.51N) also agrees well with the P velocities of the SW subregion in the depth range from 4 to 18 km. The largest differences are above 4 km and between 18 and 23 km depth. This model fits well with our S velocities of the SW subregion between 4 and 14 km depth and shows much higher velocities in other layers up to 23 km depth. The velocities extracted at 300 km along the ALP02 profile of Brückl et al. (2007) do not fit our P velocities for the E subregion very well. Only the depth interval between 6 and 18 km depth shows moderate agreement with our values. The discrepancy above 6 km depth seems to be due to the fact that the velocities were extracted at a single point along the ALP02 profile, which shows large lateral velocity variations in these layers. The same is true for the other two subregions. The S velocities determined by Živčić et al. (2000) agree well with the velocities between 16 and 23 km depth for the E, between 12 and 23 km for the NW, and between 12 and 23 km depth for the SW subregion. However, the relatively higher S velocities of the E subregion compared to the NW subregion are not consistent with their results. The model of Šumanovac et al. (2009) shows large discrepancies with our results, with higher P velocities in the depth intervals 18-23 km and 16-23 km for the E and SW subregions, respectively.

The pattern of computed P-wave station delays (Fig. 7) agrees very well with the map of sediment delay times compiled by Behm et al. (2009). The only station delays that do not agree with the sediment delay times belong to the seismic stations on the Istra peninsula. These positive station delays could be related to the relatively low velocities at shallow depths observed in the velocity models of Guidarelli et al. (2017) and Kapuralić et al. (2019), or to large-scale variations in the crust due to the limited azimuthal coverage of the observations. The travel times calculated for the southernmost three stations correspond mostly to the rays refracted at 40 km depth and above (Fig. 6). According to Stipčević et al. (2020), the Moho is deeper in this region, which means that the observed travel times are larger than the calculated ones due to longer refracted ray paths, resulting in positive station delays. According to the computed rays, the positive station delays in the east are mainly due to the rays that sampled the shallower crust with lower velocities. For the two easternmost stations, we see the opposite effect of what was described earlier. Since some computed rays were refracted below 30 km and the Moho is shallower in this region (Stipčević et al., 2020), the observed travel times are smaller than the calculated ones. The computed station delays are probably smaller, but still significantly positive due to a small number of these rays. Relatively small positive station delay of the easternmost station (MOSL) could also be due to the rays that pass through the high-density body under the Sava basin (Šumanovac et al., 2009; Šumanovac, 2010).

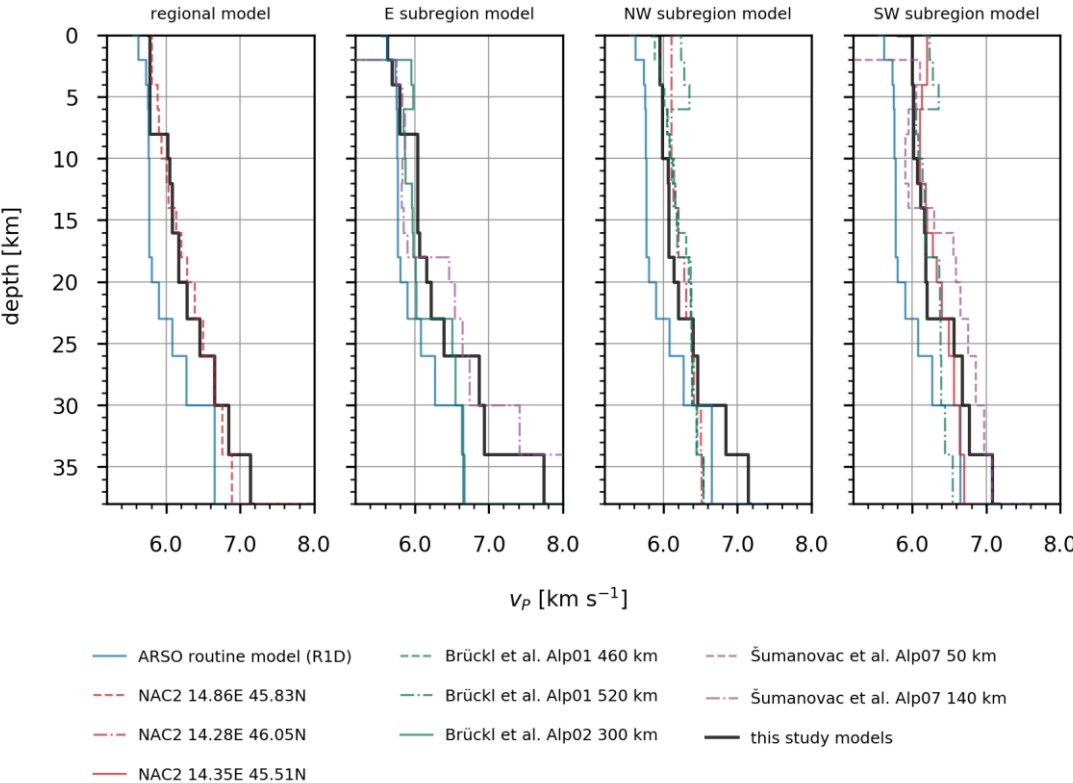

**Figure 16: Comparison of the P velocities obtained in this study (black lines) with the published velocities of Brückl et al. (2007), Magrin & Rossi (2020), Šumanovac et al. (2009) and the routine (R1D) model. Models based on the results of Magrin & Rossi (2020; NAC2) were extracted at the point closest to the respective reference seismic station. Kilometres denote the distance along the profiles in Brückl et al. (2007) and Šumanovac et al. (2009). For easier comparison, all published models were extracted by calculating the weighted average of the velocities in each layer.**

In the western part of the study area, the seismicity relocated with the MPS model (Fig. 11) is confined to depths between 0 and 20 km, which corresponds to the depths determined by Vičič et al. (2019) for western Slovenia. Towards the eastern part of the study area, the earthquake hypocenters become shallower and mostly occur below 15 km depth. The shallowing of the earthquake hypocenters is consistent with the shallowing of the Moho depth, as already suggested by Stipčević et al. (2020). Moreover, most of the seismicity in the region seems to be confined to the depths above the depth interval of the rapid P velocity increase and to the depths with relatively higher $v_P/v_S$ values. Therefore, the lower $v_P/v_S$ values and the increase in velocity gradient in the lower crust indicate a change in physical properties that prevents the occurrence of deeper earthquakes. A similar observation has already been made in several studies (Bressan et al., 2009; Bressan., 2012; Magrin & Rossi, 2020) that related the spatial distribution of seismicity in the northern part of the Adria to the changes in various

physical parameters in the crust. A look at the depth distribution of the relocated seismicity shows that about 2 percent of the earthquakes were relocated to depths between 0 and 1 km. These earthquakes occurred mostly at the periphery of our study

area (Fig. S7). Relocation using the velocity model for the corresponding subregion mitigated this problem only in the area of the SW subregion, where one event remained at a depth above 1 km and even this one directly at a quarry. Since other events in this region could not be attributed to quarries and were relocated to greater depths with the subregional model, this suggests that the shallow structure was more accurately resolved with inversion at a smaller scale, as also previously shown by Husen et al. (2011). Thus, despite a relatively poor performance in the stability tests compared to the other two

subregions, the model for the SW subregion still provides reliable earthquake locations.

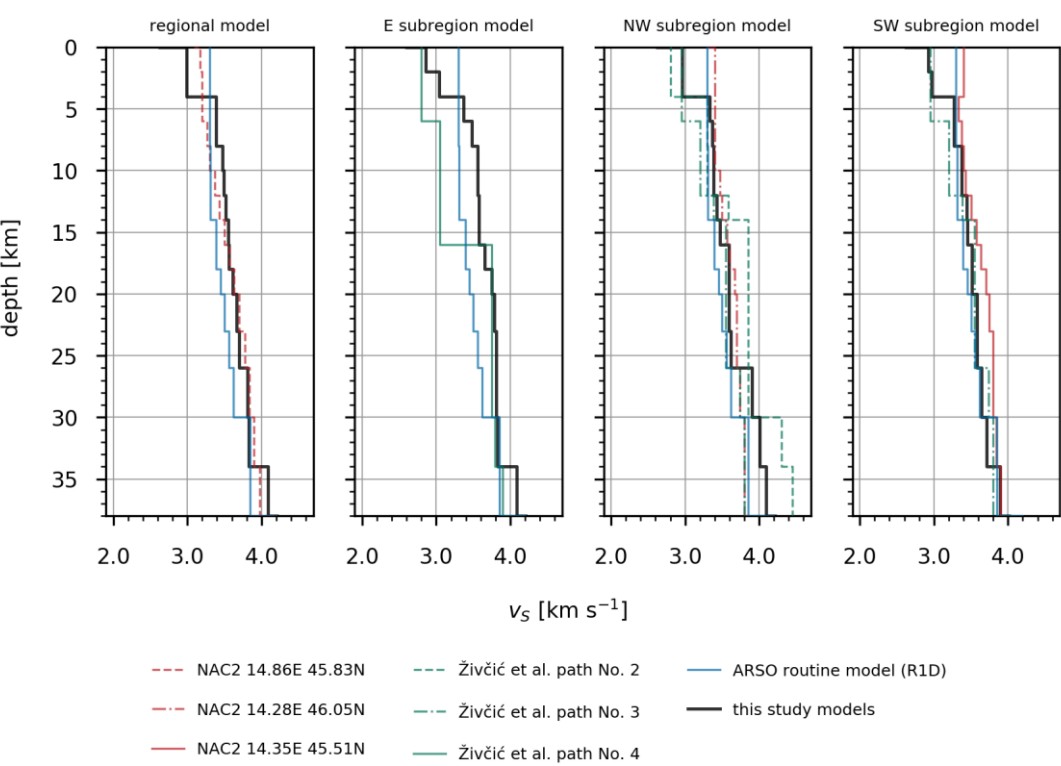

**Figure 17: Comparison of the S velocities obtained in this study (black lines) with the published models of Magrin & Rossi (2020), Živčić et al. (2000) and the routine (R1D) model. Models based on the results of Magrin & Rossi (2020; NAC2) were extracted at**
730 **the point closest to the respective reference seismic station. For easier comparison, all published models were extracted by calculating the weighted average of the velocities in each layer.**

Earthquake hypocenters in the north-west that remain at depths above 1 km after the relocation with the velocity model for the NW subregion are probably still accurate due to the high topography in this area. Such hypocenters in the east could be biased toward the surface because a 1-D velocity model cannot fully account for the local effects of deep sedimentary basins. Since 16 more earthquakes in the east were relocated above 1 km depth using the velocity model for the E subregion, two large P velocity jumps in this model computed for the unconstrained layers below 23 km depth (Figs. 13 and 16) may also bias the earthquakes with readings from distant stations. In addition, only about 3.5 percent of earthquakes in the E subregion were relocated to depths above 1 km, and most of them with depths above 0.5 km had high RMS residuals ranging from 0.37 to 0.97 s. For this reason, unconstrained depths could also result from some potentially inaccurate readings with high residuals. Due to their proximity to quarries, five of these events most likely resulted from explosions. Lower RMS residuals obtained for the subregions also indicate a better fit of the obtained velocities and better resolved station delays, as there are relatively fewer differences between the structure below the reference station and all other stations. The gap in seismicity above about 7 km depth in some parts in the west can be observed in both the relocated and routinely located seismicity. In the east, the earthquakes are in some parts already absent below about 10 km depth. Since earthquakes have occurred at comparable depths in the other parts of the study area, the apparent absence of earthquakes could be due to the relatively short time span of our dataset. If there are structural reasons for this type of depth distribution of earthquakes, we expect to find them with the computation of a 3-D velocity model.

With this study, we evaluated in detail the performance of 1-D velocity inversion, which has been shown many times to be essential for the results of LET (e.g., Kissling et al., 1994). The obtained regional and local 1-D velocity models provide reliable hypocenters of local events using first P and S arrival times. Based on the results for the subregions, the study area cannot be considered uniform in terms of seismic velocity and seismicity. This means that using only one model to locate earthquakes at the regional level may bias the hypocenters, even with the computed station delays. As can be seen from our study, the station delays cannot always account for the full effect that lateral velocity variations in the shallow crust have on travel times, especially in the case of deep sedimentary basins. Further work is needed in the study area to obtain a more reliable 1-D P and S velocity model for the Dinarides further south, where the data selection process is more challenging due to the smaller number of seismic stations. Based on the results of this and other studies (Stipčević et al., 2020) showing a highly variable $v_P/v_S$ in the study area, the P and S velocity models resulting from the combined P and S inversion provide an improvement for the relocation of the seismicity compared to the constant $v_P/v_S$ often used in studies in this region. The results of this study will also be used to compute a high-resolution 3-D velocity model that has the potential to resolve tectonic structures in the upper crust in greater detail and link tectonics to seismicity. The division into subregions allows us to further investigate the ray sampling and seismicity distribution of each subregion, which in turn allows better preparation for a 3-D tomography.

*Code and data availability.* Routine hypocenter locations were obtained using the HYPOCENTER program (Lienert & Havskov, 1995). Hypocenter-velocity inversions were performed using the shareware program VELEST (Kissling et al., 1994). Figures and maps were plotted in Python using Matplotlib (https://matplotlib.org) and Cartopy

(https://scitools.org.uk/cartopy) packages.

*Seismic bulletins and catalogues*: Earthquake information is provided by the Slovenian Environment Agency (ARSO, seismological bulletin 2004-2018 and earthquake information) and University of Zagreb (earthquake information).

*Permanent seismic networks:*

INGV Seismological Data Centre: Rete Sismica Nazionale (RSN), Istituto Nazionale di Geofisica e Vulcanologia (INGV), Italy, https://doi.org/10.13127/SD/X0FXnH7QfY, 2006.

MedNet Project Partner Institutions: Mediterranean Very Broadband Seismographic Network (MedNet), Istituto Nazionale di Geofisica e Vulcanologia (INGV), https://doi.org/10.13127/SD/fBBBtDtd6q, 1990.

OGS (Istituto Nazionale di Oceanografia e di Geofisica Sperimentale) and University of Trieste: North-East Italy Broadband Network, International Federation of Digital Seismograph Networks, https://doi.org/10.7914/SN/NI, 2002.

OGS (Istituto Nazionale di Oceanografia e di Geofisica Sperimentale): North-East Italy Seismic Network, International Federation of Digital Seismograph Networks, https://doi.org/10.7914/SN/OX, 2016.

Slovenian Environment Agency: Seismic Network of the Republic of Slovenia, International Federation of Digital

Seismograph Networks, https://doi.org/10.7914/SN/SL, 2001.

University of Zagreb: Croatian Seismograph Network, International Federation of Digital Seismograph Networks, https://doi.org/10.7914/SN/CR, 2001.

ZAMG – Zentralanstalt für Meteorologie und Geodynamik: Austrian Seismic Network, International Federation of Digital Seismograph Networks, https://doi.org/10.7914/SN/OE, 1987.


*Temporary seismic networks:*

AlpArray Seismic Network: AlpArray Seismic Network (AASN) temporary component, AlpArray Working, https://doi.org/10.12686/alparray/z3_2015, 2015.

**Team list**

The complete member list of the AlpArray Working Group:

György Hetényi, Rafael Abreu, Ivo Allegretti, Maria-Theresia Apoloner, Coralie Aubert, Simon Besançon, Maxime Bès De Berc, Götz Bokelmann, Didier Brunel, Marco Capello, Martina Čarman, Adriano Cavaliere, Jérôme Chèze, Claudio Chiarabba, John Clinton, Glenn Cougoulat, Wayne C. Crawford, Luigia Cristiano, Tibor Czifra, Ezio D'alema, Stefania Danesi, Romuald Daniel, Anke Dannowski, Iva Dasović, Anne Deschamps, Jean-Xavier Dessa, Cécile Doubre, Sven

Egdorf, Ethz-Sed Electronics Lab, Tomislav Fiket, Kasper Fischer, Wolfgang Friederich, Florian Fuchs, Sigward Funke, Domenico Giardini, Aladino Govoni, Zoltán Gráczer, Gidera Gröschl, Stefan Heimers, Ben Heit, Davorka Herak, Marijan Herak, Johann Huber, Dejan Jarić, Petr Jedlička, Yan Jia, Hélène Jund, Edi Kissling, Stefan Klingen, Bernhard Klotz, Petr Kolínský, Heidrun Kopp, Michael Korn, Josef Kotek, Lothar Kühne, Krešo Kuk, Dietrich Lange, Jürgen Loos, Sara Lovati, Deny Malengros, Lucia Margheriti, Christophe Maron, Xavier Martin, Marco Massa, Francesco Mazzarini, Thomas Meier,

Laurent Métral, Irene Molinari, Milena Moretti, Anna Nardi, Jurij Pahor, Anne Paul, Catherine Péquegnat, Daniel Petersen, Damiano Pesaresi, Davide Piccinini, Claudia Piromallo, Thomas Plenefisch, Jaroslava Plomerová, Silvia Pondrelli, Snježan Prevolnik, Roman Racine, Marc Régnier, Miriam Reiss, Joachim Ritter, Georg Rümpker, Simone Salimbeni, Marco Santulin, Werner Scherer, Sven Schippkus, Detlef Schulte-Kortnack, Vesna Šipka, Stefano Solarino, Daniele Spallarossa, Kathrin Spieker, Josip Stipčević, Angelo Strollo, Bálint Süle, Gyöngyvér Szanyi, Eszter Szűcs, Christine Thomas, Martin

Thorwart, Frederik Tilmann, Stefan Ueding, Massimiliano Vallocchia, Luděk Vecsey, René Voigt, Joachim Wassermann, Zoltán Wéber, Christian Weidle, Viktor Wesztergom, Gauthier Weyland, Stefan Wiemer, Felix Wolf, David Wolyniec, Thomas Zieke, Mladen Živčić, Helena Žlebčíková.

*Author contributions.* G.R., J.S., M.Ž., and A.G. conceptualised the study. G.R. performed the inversion, programmed

supporting algorithms, developed the tests, prepared the figures, and wrote the original draft of the manuscript. G.R., J.S., M.Ž., and M.H. validated the results. G.R., J.S., M.Ž., M.H., and A.G. curated the data and were involved in investigation. A.G. supervised the study and acquired funding. All co-authors discussed the methods and results and reviewed and edited the manuscript. The AlpArray Working Group provided access to seismic data from the temporary stations.

*Competing interests.* The authors declare that they have no conflict of interest.

*Acknowledgments.* This study was carried out with the support of the Research Program P1-0011 and the Young Researcher grant (1000-21-0510), funded by the Slovenian Research Agency. This work has also been supported in part by the Croatian Science Foundation under the Project No. IP-2020-02-3960. Many thanks to Edi Kissling and Matteo Bagagli for providing

the latest version of the VELEST code and for discussion. We also thank Emanuel Kästle for providing the base geology and

tectonics layers. We are grateful to Giuliana Rossi and an anonymous reviewer whose constructive comments significantly improved this paper.

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
