# Peer review of "1-D velocity structure modelling of the Earth's Crust in the NW Dinarides"

_Solid Earth, 2021_

## Author Comment (AC2)

**Supplement - Relocation figures**

[Figure]

**Figure S1:** Proposed replacement for Figure 11 with lines showing the shift of relocated hypocenters relative to the routine locations.

[Figure]

**Figure S2:** Proposed replacement for Figure 11 with open circles instead of filled ones for the routine locations.

[Figure]

**Figure S3:** Final replacement for Figure 11 with smaller markers on the main map and relocated hypocenters put on the top of the routine hypocenters.

---

## Author Response (AR1)

**Reply to RC1**

First of all thank you very much for your time invested in this review and the suggestions which we think are very constructive.

Regarding the **first main point** about S wave readings. We took your suggestion into account and calculated the new models. We additionally calculated the S velocity model, which we used, along with the P velocity model, as an initial model in the combined inversion for both P and S velocities. Furthermore, we took the P and S models and relocated hypocenters from the regional inversion, made earthquake selection for each subregion and did P and S inversion for each of them. For each additional inversion we did all the tests as already described in the manuscript. We included the results of the newly done inversions in the manuscript and the supplement. Judging by relocations of earthquakes and blasts, we estimate that the hypocentral depths were improved using P and S models. The recovered models show a change in velocity but retained their main features as in the models using only P phase. The results and discussion sections were rewritten accordingly. Minor changes were also made to the other sections.

**Second point** was about the selection criteria, namely reading classes and the absolute residual of each observation. The possible exclusion of longer ray paths with relatively large residuals, but otherwise good quality picks is the exact reason why we did not filter for individual residuals and only used an event RMS as a selection criterion. On top of that, we can not improve the sampling of the lower crust any further, since the earthquakes in this area rarely occur below 20 km depth and the investigated region is too small for rays to penetrate deeper. Thus, adding longer ray paths would only introduce more subvertical ray paths into deeper layers, which would not improve the constraints on velocity there.

We excluded the reading class of 3 as there are very few readings with this class assigned (see Table S1 in the revised supplement). Even adding all of them, they would not contribute much, and we risk the potential to make the inversion unstable.

**Minor points.**

**Reviewer:** The tectonic setting should be shortened. Many details are not useful in this type of study that is focused on the 1D structure of the region. I find more interesting to report, in a simple and schematic view (for example in figure as an additional inset), the values of crustal thickness as reported by previous studies.

In the tectonic setting we removed the following text.

Lines 125-126: ". The ongoing convergence between the Adria and the Eurasia could be compensated here by underthrusting of the Adriatic mantle under the Pannonian mantle"

We find all other information relevant for our study and necessary to acquaint the reader with the study area. Since we are not dealing with crustal depth and are not using those values in our calculation, we only report crustal thickness in the text.

**Reviewer:** At lines 196-198: "The 1-D model approaches the average of the 3-D velocity model blocks, weighted by the total ray length in each block": I find that this phrase is misleading since the block discretization is more appropriate for a tomographic approach. I suggest to remove it and to continue with ". In other words, the layer velocities of a 1-D velocity model approximate the average velocity of a 3-D velocity model in the same depth interval"

We completely agree with the point you made regarding the velocity blocks (lines 196-198) and made the proposed change in the revised manuscript.

**Reviewer:** At lines 211-212 we have "The differences between the adjacent layers were kept as small as possible to ensure stability during the inversion": could you explain better, difference of thickness, velocity?

We added the text about a difference in thickness among adjacent layers. We hope it is more clear now.

"The difference in thickness between the adjacent layers was kept as small as possible to ensure stability during the inversion."

**Reviewer:** Cell division of earthquakes: the crustal volume discretization was not performed along *Z* but only along *X* and *Y*. Then, along *Z* you check the inter distance among hypocenters. Why this choice? Why not discretize along *Z*?

By only discretizing along Z, we still obtained clusters in space, which is something we try to avoid in such inversions, because adding similar ray paths contribute nothing to the inversion (see Supplement), and present only a potential for instabilities to occur during the inversion. Therefore, we support the approach of minimum distance between hypocenters as being superior.

**Reviewer:** I think that an important parameter that must enter to defined the score of one event for a cell hierarchy is the distance of the first station that it is used to locate the event.

We agree that this parameter could also be used to select earthquakes for the inversion, but do not think that it would have much influence on the results of the inversion. Our station network is dense and we expect every earthquake inside the perimeter of the stations to be relatively close to at least one station. Besides, this would mean that we could lose some important earthquakes with more quality observations. Furthermore, we do not want to introduce too many selection criteria tied to the hypocentral locations, only the most important ones, because we do not know how good the location of earthquakes really are.

**Reviewer:** Figure 11 should be improved: the comparison between ARSO and M2 locations is difficult to be appreciated since circles overlap. I suggest to remove the gray circles. I suggest drawing a thin line to connect the M2 and ARSO location, then only the M2 circle should be drawn.

Thank you for your suggestion on Fig. 11. We took it into account and prepared the figures shown below (Figs. 1, 2, and 3). Unfortunately, none of the new figures show a shift in

hypocenters clearly due to the large number of earthquakes, especially in clusters. Our intention by adding the routine locations on a map was to show that hypocenters actually shifted, but because of a large number of them it is impossible to show by how much. Because we want to emphasize the relocations, we decreased the size of the markers on the main map and put the relocated hypocenters on top of the routine ones (Fig. 3). Shift in depth of the hypocenters can already be appreciated by looking at the panel with the depth distribution of both the routine locations and relocations. Additionally, we added the statistics of the relocation into the text.